# Druggable Molecular Networks in *BRCA1/BRCA2*-Mutated Breast Cancer

**DOI:** 10.3390/biology14030253

**Published:** 2025-03-02

**Authors:** Francesca Pia Carbone, Pietro Ancona, Stefano Volinia, Anna Terrazzan, Nicoletta Bianchi

**Affiliations:** 1Department of Translational Medicine, University of Ferrara, 44121 Ferrara, Italy; francescapia.carbone@unife.it (F.P.C.); pietro.ancona@unife.it (P.A.); s.volinia@unife.it (S.V.); nicoletta.bianchi@unife.it (N.B.); 2Genomics Core Facility, Centre of New Technologies, University of Warsaw, 02-097 Warsaw, Poland; 3Laboratory for Technologies of Advanced Therapies (LTTA), 44121 Ferrara, Italy

**Keywords:** BRCA1, BRCA2, mutations, breast cancer, phytoestrogens, PARPi, pathway, drugs

## Abstract

The present study is based on an adherent analysis of the literature focused on genes deregulated by *BRCA1* and *BRCA2* mutations in breast cancer and modulated by treatments. Using the list of genes emerging from these investigations, we created networks and identified pathways useful for future targeted therapies.

## 1. Introduction

### 1.1. Breast Cancer: Overview

Breast cancer (BC) is the most widespread malignancy in women (25% of all cancers), accounting for more than 10% of new cancer diagnoses each year, and it is the second-leading cause of cancer death among women worldwide [1]. The pathology occurs when breast cells in the basal layer, inside the lobules and/or milk ducts, start to grow out of control. There are three distinct types of breast neoplasia. Ductal carcinoma in situ (DCIS) arises in epithelial cells forming the breast ducts. Several studies suggest that at least one third of DCIS cases will eventually progress to invasive cancer if left untreated. Lobular carcinoma in situ (LCIS) develops in milk producing tissue (the functional part of the breast gland), conferring an increased risk of developing invasive cancer. Cancer is considered invasive when the tumoral cells cross the basal membrane and infiltrate the surroundings. However, most BC are invasive or infiltrating, and the prognosis is dependent on the stage of the disease [2]. Doctors are engaged in determining the most suitable treatment approach for each patient following diagnosis. However, if BC is left untreated, it has the potential to infiltrate nearby tissues and spread to other districts of the body. BC is categorized into various stages using the Tumor, Node, Metastasis (TNM) Classification of Malignant Tumor from the AJCC/UICC staging system [3]. Four molecular phenotypes (luminal A, luminal B, HER2^+^, and triple-negative breast cancer (TNBC)) are characterized by the presence of estrogen (ER^+^) and progesterone receptors (PR^+^), as well as human epidermal growth factor receptor 2 (HER2^+/neu^) [4]. TNBCs, characterized by worst prognosis, are the less frequently diagnosed BC (10–20%) [1]. They were previously divided into four subgroups [5], further categorized by whole-genome sequencing [6] or transcriptomic analysis [7].

Luminal A and B tumors, expressing ER and/or PR, typically respond to chemotherapy and endocrine treatments. ER modulators, like tamoxifen and fulvestrant, are commonly used both for pre- and post-menopausal patients, though some individuals develop resistance or become unresponsive. To improve the prognosis for hormone receptor-positive patients and their treatment tolerability, new inhibitors of Cyclin Dependent Kinase 4 (CDK4), Cyclin Dependent Kinase 6 (CDK6), aromatase, or combinations of them [8] are used. Concerning HER2 tumors, various anti-HER2 drugs have been developed, including monoclonal antibodies, such as trastuzumab and pertuzumab and lapatinib. These therapies are often combined with chemotherapeutic agents, such as taxane, docetaxel, and paclitaxel, as recommended by the American Society of Clinical Oncology (ASCO) and the National Comprehensive Cancer Network (NCCN) guidelines [9]. Despite the availability of targeted approaches, HER2^+^ tumors generally have a poorer prognosis compared to luminal BC due to drug resistance onset [10]. Currently, patients with advanced and metastatic BC, often presenting HER2 enrichment or TNBC phenotypes, undergo a neoadjuvant regimen, consisting of chemo- or endocrine therapy followed by surgical tumor removal, to prevent spread of metastasis [11].

In recent years, immunotherapy has emerged as a significant advancement in cancer treatment. This approach activates immune system cells to target tumor neo-antigen molecules, which are exposed to cancerous cells due to genetic aberrations and serve as selective targets for immune system therapies [12]. Immunotherapy encompasses various methods, including vaccines, monoclonal antibodies, and checkpoint inhibitors, which enhance or restore the immune system’s ability to fight cancerous cells [13]. These techniques have provided a viable alternative to chemotherapy, particularly for highly immunogenic cancers, such as lung, renal, melanoma, and prostate cancer. Although BC was long considered immunologically “cold”, recent studies revealed a consistent expression of Programmed cell death protein 1 (PD-1) and Programmed death-ligand 1 (PD-L1) in BC stem cells associated with epithelial–mesenchymal transition (EMT) [14]. Also, the TNBC phenotype exhibits a high mutational burden, resulting in substantial tumoral neoantigen expression. This subtype is characterized by high concentrations of infiltrating T-cells, suggesting that TNBC patients, especially those with the Luminal Androgen Receptor subtype, may benefit from immunotherapy [5]. Indeed, immune checkpoints, like PD-1/PD-L1 and Cytotoxic T-Lymphocyte Antigen 4, are highly expressed in TNBC (20–30% of cases) correlating with aggressiveness and therapy resistance. These findings have prompted researchers to investigate molecules targeting immune checkpoints to treat TNBC. In this context, pembrolizumab and atezolizumab function as immunotherapy drugs [15]. Pembrolizumab targets with the PD-1 receptor, which prevents PD-L1 and PD-L2 immunosuppressive effects, while atezolizumab, targeting PD-L1, prevents the PD-1 response. This is crucial to help the immune system recognize and destroy cancer cells. Both drugs have been tested in metastatic conditions, and atezolizumab is FDA-approved for PD-L1^+^ TNBCs [16].

In particular, a KEYNOTE-522 randomized, double-blind, phase III trial evaluated the combination of carboplatin or paclitaxel with or without pembrolizumab, followed by doxorubicin and cyclophosphamide with or without pembrolizumab in patients with stage II or III TNBC. The results after 15.5 months showed that the addition of pembrolizumab improved pathologic complete response (pCR) rates. The percentage of patients with a pCR in the pembrolizumab plus neoadjuvant chemotherapy group was 64.8% (260 out of 401 patients) *versus* 51.2% in the placebo plus neoadjuvant chemotherapy group (103 out of 201 patients) [15].

Another trial, the IMpassion031 randomized, double-blind, phase III neoadjuvant treatment trial, compared atezolizumab *versus* placebo combined with nab-paclitaxel, followed by doxorubicin and cyclophosphamide in TNBC patients. The data showed that atezolizumab improved pCR rate irrespective of patients PD-L1 status; in fact, 95 out of 165 patients treated with atezolizumab plus chemotherapy had pCR. Instead, 69 out of 168 patients had pCR in the placebo plus chemotherapy [16]. These findings highlight that immune checkpoint inhibitors (ICIs) added to chemotherapy in TNBC may augment pCR, albeit long-term outcomes and toxicity in patients receiving these drugs in the neoadjuvant setting are still undergoing evaluation. Hitherto, clinical trials showed inconsistent results, with only a small proportion of BC patients responding favorably to immunotherapies [17,18].

BCs are extremely heterogeneous, owing to differences in genomic, epigenomic, transcriptomic, and proteomic characteristics of the cells. The high heterogeneity influences tumor properties and therapeutic response; therefore, personalized and tailored approaches are becoming the standard in dealing with this pathology [19,20].

### 1.2. Genetic Causes of BC

The tumor suppressor p53 is a major transcription factor involved in the regulation of several cellular functions. In cancer, p53 inhibits cell proliferation in response to multiple stimuli, including DNA damage, food deprivation, hypoxia, and hyperproliferative signals, therefore inhibiting tumor growth, leading to his role of “guardian of genome”. Notwithstanding, *TP53* is one of the most frequently mutated genes in BC (about 30%) [21]; the encoded protein works as a tetramer and each monomer consists of several domains: an N-terminal transactivation domain (TAD) subdivided into two regions (TAD1 and TAD2), a proline-rich domain (PRD), a core DNA-binding domain (DBD), a tetramerization domain (TD), and a C-terminal regulatory domain (RD) [22].

TAD are required for p53 binding to cofactors and for the p53-mediated suppression of tumorigenesis in response to stress. Still, each transactivation domain provides p53 with cofactor binding selectivity, altering the cells’ overall response to a given stress. TAD is also responsible for the binding of MDM2, a p53 primary negative regulator, while TAD1 loss, during the DNA damage response (DDR), together with *Ras* oncogene expression, results in the abolition of p53 response, serving as a p53-null cell [23].

TAD2 disruption instead preserves similar wild type capabilities, such as the ability to activate p53 target genes coding P21, the Bcl-2-associated X protein (BAX), NOXA, and the p53 upregulated modulator of apoptosis, PUMA, as well as cell cycle arrest and death. The simultaneous loss of TAD1 and TAD2 abolishes p53 activity, resulting in a p53-null response in vivo, including the inability to induce senescence and the susceptibility to tumor lesions. These findings demonstrate that TADs provide the protein p53 with a level of specificity for gene transcription; whether this holds consistent in response to other stress stimuli or in vivo cancer models must be investigated [24].

Another crucial domain for the ability of p53 to bind DNA is the PRD, which has been related to the suppression of the colony formation of tumor cells in vitro. However, its deletion has not been linked with an interruption of the suppression function. The consequences of deletion are p53 nuclear export, leading to susceptibility to ubiquitination, MDM2 degradation, and reduced p53 transcriptional activity, prompting deleterious implications. The PRD role was also demonstrated in vivo, and its deletion compromised thymocytes to undergo apoptosis upon irradiation [25,26].

The DBD of p53 allows its function as a transcription factor by recognizing the p53 responsive element (RE), distinguished by a specific DNA sequence. At least two copies of the RE are essential for a gene to be transcriptionally regulated by p53 [27].

The TD is responsible for oligomerization as a tetramer of four p53 proteins, allowing it to achieve the appropriate conformation to bind DNA for sequence recognition. Loss in this area impairs DNA binding and interferes with protein interactions. Moreover, p53 is crucial for post-translational modification such as phosphorylation and ubiquitination [28].

The RD is responsible for inhibiting the p53 DNA binding domain until stress signals trigger changes. This causes a post-translational modification in RD, such as acetylation and phosphorylation, allowing p53 transition from inactive to active conformation and enabling DBD to bind the RE. After the modification, RD can no longer prevent the p53 DNA binding domain [29].

Mutations in *TP53* are involved in DNA damage associated with apoptosis and can confer resistance to various treatments by deregulating apoptosis pathways.

Besides the previously mentioned gene, whose mutation can result in BC development, other genes are responsible for BC development; in particular, about 5–10% of all BC cases are hereditary and frequently related to mutations in two tumor suppressor genes, *BRCA1* and *BRCA2,* and to the partner and localizer of BRCA2 (PALB2) [30].

The mutations in *BRCA1/2* expose women, and eventually men, to a statistically significant higher risk of developing BC, which often has an early onset compared to that of sporadic ones [31]. *BRCA1* and *BRCA2* are mostly attributed to germline pathogenic variants (PV). Around 80% of *BRCA1* mutation carriers develop BC exhibiting the TNBC subtype. Germline PV mutations in *BRCA2* are rarely associated with TNBC (2–16%) but are more commonly reported in association with ER^+^ BC [32]. From the age of 70 years, the accumulation of mutations increases the risk of BC from 33 to 58% [33,34], and hereditary BCs are thought to be linked to *PALB2*, a gene with a moderate to high risk. In addition, family history and other environmental factors influence the risk associated with a *PALB2* PV. The cumulative risk for women without a family history of BC is 33%, while the cumulative risk for those with two or more family members affected by BC is 58%. The phenotypic features of breast malignancies in people with *PALB2* mutations are comparable to those of tumors with *BRCA1/2* mutations [35]. Interestingly, cancer phenotypes induced by a single mutation can vary, which is one of the most harmful aspects.

## 2. Materials and Methods

In order to analyze the most recent and promising literature, we generated a query based on a combination of the above inclusion criteria: ((“genes, BRCA1”[MeSH Terms] OR “BRCA2 siRNA”[All Fields] AND “breast cancer”[All Fields]) AND (“siRNA”[All Fields] OR “DNA damage”[All Fields] OR “apoptosis”[All Fields] OR “deficiency”[All Fields] OR “inhibition”[All Fields] OR “expression of genes”[All Fields] OR “resistance”[All Fields]) NOT (Review[pt]) NOT (Meta-analysis[pt]) NOT (“Systematic Review”[pt])). This has returned 282 articles, a high number, which, however, guaranteed that we would maintain the important inclusion criteria essential to us, mainly concerning the connections among treatments in relationship to one/more gene/genes, which are modulated by the *BRCA* mutations, necessary to reach the goal of this review. After a manual curation of the literature, only 14 articles were considered useful for further insights due to the characteristics required. We chose articles including features such as the mutations of *BRCA1* or *BRCA2*, genes affected by these mutations and their up or downregulation, the cell type used for validation, and, finally, the presence of an associated treatment. In this context, we highlighted the correlation between therapies and gene expression. Then, we investigated each article to group the ones with the single mutation in *BRCA1* and those with mutations in both *BRCA* genes. We distinguished the papers according to the positive or negative effects of the mutation. Moreover, we separated the direct or indirect influence of BRCA1 and BRCA2 on the target genes involved.

Summarizing, we constructed networks by using Cytoscape 3.10.2 in order to determine which genes are affected by *BRCA* mutations, distinguishing the networks based on the positive or negative impact on each single gene.

The framework is schematized in Figure 1.

The genes obtained from the query were listed to create the networks. Their nomenclature was standardized by using HUGO Gene Nomenclature Committee (HGNC). To clarify the inhibition or activation effects of *BRCA* mutation on these genes, different arrowheads were used, such as flat ones to represent inhibition and pointed ones to represent the activation. Moreover, we defined primary and secondary order interactions based on different lines shapes, outlining dashed lines for secondary interactions whilst using solid lines for primary interactions. An additional trait of this network is the utilization of different colors, as described in the legend of the figures, in order to distinguish the experimental therapies associated with genes modulated by *BRCA1/BRCA2* mutations.

Finally, the pathway analysis was performed by using the WEB-based Gene SeT Analysis Toolkit (WebGestalt 2024, https://www.webgestalt.org/, accessed on 20 December 2024). We conducted an over-representation analysis (ORA), selecting the Kyoto Encyclopedia of Genes and Genomes KEGG Pathways (https://www.genome.jp/kegg/pathway.html, accessed on 20 December 2024) as a functional database. The sources and the list of genes are reported in Appendix A.

The resulting modulation of the involved genes could represent a valuable resource to explore new targets for drug design and repurposing as chemopreventive agents. Hence, our investigation sheds light on potential target genes, which have been already shown to be affected by molecules and compounds.

This review aims to analyze the literature based on *BRCA*-mutated models, mostly to verify which genes are modulated or up or downregulated by the *BRCA* gene mutation. In this work, we only selected the studies that included all of our criteria, discovering several genes that are not commonly associated with *BRCA* mutation in the literature. Furthermore, our analysis identified genes regulated by treatment and compounds used in combination therapies. Despite the lack of treatment, we included two single-cell-based studies that highlighted not only some key genes but also features such as the induction of the basal-mesenchymal transition of the luminal progenitors responsible for breast cancer development in mutation carriers. Besides that, luminal progenitors have also been investigated for their ability to develop alveolar differentiation during premalignancy in *BRCA1* mutated models.

## 3. Relationship Between *BRCA1/2* Mutations, Gene Expression, and Biological Processes

In this review, we highlighted the most promising studies in the field and examined the relationship between *BRCA1* and *BRCA2* mutations and their impact on the expression of other genes. The goal is to identify the genes mainly involved in *BRCA*-mutated BCs, mostly linked to inherited breast malignancies.

### 3.1. BRCA Mutations Interfere with Homologous Recombination and Cell Cycle

*BRCA1* and *BRCA2* transduce two fundamental members of the protein complex involved in the homologous recombination (HR), ensuring an accurate repair of DNA double-strand breaks (DSBs). Their activation is linked to phosphorylation, which can occur in different aminoacidic residues, depending on the stimuli and, therefore, on the various kinases involved [36]. BRCA1/BRCA2 collaborate with Rad51, a crucial protein for HR during mitosis, meiosis, and DNA repair. Rad51 forms nucleoprotein single-strand filaments, enabling homologous DNA strand invasion and duplex formation during the recombination. BRCA1 and Rad51 work together in repair processes and DSB detection [36]. In this context, the main BRCA1 function involves the regulation of Mre11 single-strand nuclease, which creates single-strand DNA at break sites [37], while the role of BRCA2 in DSB repair is more direct. Indeed, BRCA2 and Rad51 form complexes that exist in active or inactive states. The inactive state prevents Rad51 interaction with single-stranded DNA, while the active form guides Rad51 to DNA-damaged sites, facilitating nucleoprotein filament formation. The transition between these states is regulated by post-translational modifications, depending on protein phosphorylation [36].

Consequently, BRCA1/BRCA2 are crucial also in cell cycle checkpoints, which stop DNA replication when breaks occur. BRCA1 leads to defective DNA damage repair, abnormal centrosome duplication, G2/M cell cycle checkpoint defects, growth retardation, increased apoptosis, genetic instability, and tumorigenesis, whereas the role of BRCA2 in cell cycle regulation is not fully understood, although evidence suggests its involvement [38,39,40].

Yarden et al. made a significant contribution to this topic, demonstrating that BRCA1 regulates the expression, phosphorylation, and cellular localization of checkpoint kinase 1 (Chk1), a serine/threonine-specific protein kinase, which is a known regulator of the G2/M cell cycle checkpoint [41,42]. The DDR cascade begins with upstream kinases, such as the ataxia-telangiectasia-mutated (ATM) and ATM-related kinases (ATR), which phosphorylate the DNA damage sensors, Chk1 and Chk2, who play pivotal roles [43,44,45].

A study by Brodie and Deng [46] demonstrated that *BRCA1*-deficient mice showed a loss of heterozygosity in some tumor suppressor genes and an amplification of genomic regions containing oncogenes. *BRCA1* knockout resulted in the overexpression of specific genes, such as Erb-B2 receptor tyrosine kinase 2 (*ErbB2*), the proto-oncogene bHLH transcription factor (*c-Myc*), and Cyclin D1, which were also observed in cell lines derived from primary tumors.

The identification of chromosomal abnormalities, including structural and copy number variations, suggests that genomic instability, arising from *BRCA1* loss, leads to mutations in other genes. *ErbB2* is detected in less than 25% of *BRCA1*-mutated tumors. It encodes a glycoprotein member of the epidermal growth factor receptor (EGFR) family, lacking the ligand binding domain that affects the bond to growth factors [47]. Immunohistochemical and Western blot analysis on primary tumors and cell lines from *BRCA1* knockout mice demonstrated an *ErbB2* overexpression; furthermore, these models expressed Cyclin D1 at high levels [48].

We underline that half of the *BRCA1*-mutated tumors were positive for Cyclin D1 expression [49] and most of them are p27^+^.

Brodie et al. [50] confirmed the overexpression of these genes in all tumors, albeit at different levels, together with *c-Myc*. Moreover, in mouse mammary epithelium, the overexpression of *c-Myc* and *ErbB2* is sufficient alone to induce tumor formation [48,49,50,51,52]. These observations found in sporadic forms of BC are instead frequent in *BRCA1*-mutated tumors.

Under normal conditions, the interaction between p53 and full-length BRCA1 has been demonstrated, responsible for the p21 upregulation, driving the arrest of the cell cycle in G1/S [53,54,55]. The complete absence of BRCA1 in embryos exhibited severe developmental delays and cellular proliferation defects, which can be partially rescued by loss-of-function mutations in p53 or p21 [56]. Given that p53 regulates the G1/S cell cycle checkpoint via p21 activation, these data suggest that the loss of BRCA1 may activate a p53/p21-mediated cell cycle checkpoint, potentially leading to cell death.

BCs display mutations in several BRCA1 interacting proteins (BIPs), including BRCA2, ATM, and BRCA1-associated RING domain 1 (BARD1). Carcinogenesis depends on the interactions between BRCA1 and BIP; therefore, BIP mutations should be observed in *BRCA1* familial cancers. In this context, p53 is the most promising “BIP” linked to BRCA1 driving carcinogenesis [57]. Early-onset BC, known as Li–Fraumeni syndrome, is caused by germline mutations in p53 [58]. *BRCA1* familial BCs have a much higher incidence than sporadic ones. However, it is not clear whether this is simply a consequence of the genetic instability associated with the *BRCA1* mutation or if it is promoted by factors accelerating tumor formation. In an attempt to clarify the consequences of these mutations, Xu et al. introduced a p53-null mutation into the BRCA1 conditional-mutant mice, demonstrating that when both genes were mutated, mammary tumor formation was dramatically accelerated [59].

What emerges are not only the roles of BRCA1 and BRCA2 in DNA repair and cell cycle regulation, but also their relevance when HR occurs. For this reason, the mutation of the two genes is strongly involved in genomic instability, leading to the overexpression of oncogenes and the loss of cell cycle control and, consequently, to cancer development.

### 3.2. BRCA1/2 Modulate Transcription of Specific Genes

BRCA1 plays a role in modulating various genes involved in DDR also through transcriptional regulation mechanisms. BRCA1 interacts with histone deacetylase complexes and is associated with DNA polymerase II holoenzyme, directly engaging with RNA helicase A [46], as well as with c-Myc and p53. C-Myc is typically considered a transcriptional activator and BRCA1 suppresses its transcription along with cancer transformation; concerning the relationship between BRCA1 and p53, the wild type BRCA1 protein enhances the transcription of p53-responsive genes at their promoters, coding p21 and BAX. In contrast, *BRCA1*-deficient mutants do not exhibit this effect. While p53 can influence the expression of numerous pro-apoptotic genes, the increased expression of *BRCA1* does not induce apoptosis in most of the cell lines examined. This may be partly attributed to BRCA1 activity, which selectively upregulates p53-dependent genes involved in DNA repair and cell cycle arrest [60].

The BRCA2 role is less clearly defined than that of BRCA1. However, some evidence suggests a connection between BRCA2 and other transcription-related proteins, such as Smad3. Both Smad3 and BRCA2 contain transcription-activation domains and exhibit functional and physical interactions. Previous studies have demonstrated that BRCA2 forms a complex with Smad3, both in vitro and in vivo, working synergistically to regulate gene transcription [61].

### 3.3. Differences Between BRCA1 and BRCA2 Mutations in Trascriptional Regulation and Dysregulation of Cell Cycle

Deregulation of cell cycle machinery is a common finding in BC and is frequently secondary to alterations in the proteins controlling the G1/S transition, including cyclins, CDKs, and cyclin-dependent kinase inhibitors (CDKIs) [62]. Despite their importance in tumor development, there is limited data available on the expression of cell cycle-engaged proteins in *BRCA1/2*-mutated BC. In addition, cell cycle dysregulation due to *BRCA1* or *BRCA2* mutations could involve different patterns [63].

Palacios et al. [64] analyzed the expression of 37 immunohistochemical markers and the amplification status of four genes in primary infiltrating ductal carcinomas from *BRCA1/2* mutation carriers and age-matched control subjects. The results showed that the *BRCA1*-mutated tumors were characterized by hormone receptor negativity and a high frequency of p53^+^, whereas *BRCA2* showed the opposite.

Another key distinction between BRCA1 and BRCA2 in terms of cell cycle protein expression involves type D cyclins (D1, D3), their associated CDK such as CDK4, and the CDKIs p16, p21, and p27, which showed a lower expression in BRCA1 compared to BRCA2 carcinomas [65]. In fact, the ER^+^/p53^−^ phenotype, peculiar to most BRCA2 carcinomas, exhibited a reduced Ki67 expression and elevated levels of cell cycle proteins [64]. Cyclin D1 expression, which has an estrogen induction, was less frequent in *BRCA1* than in *BRCA2*-mutated carcinomas, with gene amplification rates of 18% in *BRCA1* and 60% in *BRCA2*. However, the limited number of viable cases prevents a definitive connection [65]. This observation aligns with expectations, given that cyclin D1 is estrogen-induced and its association with ER^+^ status has been established in BC [32]. While these findings might imply the deregulation of the p53 pathway by alternative mechanisms in the two genotypes, the associated expression of p21 suggests the functional preservation of this pathway in *BRCA2*-mutated carcinomas [64,65]. Regarding apoptotic markers, BCL2 overexpression was observed in *BRCA2*-mutated carriers, which correlates well with ER^+^ status. In contrast, *BRCA1* cancers displayed higher levels of caspase 3. The latter is a cytosolic enzyme activated only in cells committed to apoptosis, showing a strong correspondence with the morphological assessment [64].

*PALB2* is another significant gene and risk factor in hereditary BC [66,67]. For example, Fanconi anemia is brought on by a bi-allelic mutation in *PALB2*, which also makes children more susceptible to pediatric cancers, such as acute myeloid leukemia, Wilm’s tumor, and medulloblastoma [68]; on the other hand, mono-allelic mutation promotes ovarian cancer in women, prostate cancer in males, and familial breast and pancreatic cancers [69,70].

## 4. Targeted Therapies

### 4.1. Genes Involved in BC Resistance to Treatment

*TP53* mutations are strongly associated with chemoresistance, particularly to anthracyclines, such as epirubicin and doxorubicin [71,72]. In particular, resistance to epirubicin has been associated with mutations affecting the L2/L3 zinc binding domains of p53, reported in locally advanced BCs [72]. *TP53* mutations were also found to predict poor prognosis and early relapse in BC patients treated with doxorubicin [68]. Interestingly, while *TP53* mutations are associated with resistance to many therapies, in some cases they may confer sensitivity to specific anthracyclines. This hypothesis is confirmed by a gene set enrichment analysis performed on an integration of transcriptome data, genetic alterations, and clinical data by Liu et al. [73]. The results from the analysis showed an association between the *TP53* signaling pathway and anthracycline sensitivity. Indeed, *TP53* mutations were more common in anthracycline-sensitive tumors, highlighting the complex role of *TP53* in the treatment response, while *CDH1* mutation was presented in the anthracycline resistant group. To summarize, *TP53* mutations play an important role in BC drug resistance by altering normal apoptotic responses to DNA damage generated by chemotherapy. These effects may be context-dependent, so *TP53* status does not always predict treatment outcomes.

Evaluating *TP53* along with other genes, such as checkpoint kinase 2, may provide a more comprehensive picture of potential drug resistance mechanisms [72]. Further studies are needed to fully understand how the different *TP53* mutations impact treatment efficacy across various subtypes and therapeutic regimens.

Approximately 70% of breast tumors express ERα and rely on estrogen binding to promote carcinogenesis. Tamoxifen, an anti-estrogen, is commonly used to treat ERα^+^ tumors because it binds to the ligand-binding domain causing a conformational shift in the receptor, which initially reduces the ability of ERα to interact with the target genes. Tamoxifen-induced oxidative damage is crucial for therapeutic efficacy, along with the tamoxifen- ERα interaction [29,74]. By increasing reactive oxygen species (ROS) and inhibiting ROS activity, anti-estrogen sensitivity decreased [75]. Despite the potential of Proteolysis-targeting chimeras (PROTACs), a promising new class of drugs capable of degrading cellular proteins of interest, such as oncogene products, resistance to PROTACs has been developed. This kind of degradation treatment has highlighted the essential role of the drug efflux pump MDR1 (multidrug resistance 1) encoded by ABCB1, a member of the human ATP-binding cassette (ABC) transporter. When overexpressed, MDR1 has been shown to confer resistance to various anticancer drugs, including chemotherapy agents, kinase inhibitors, and other targeted therapies [76].

Endocrine resistance could evolve from many mechanisms, including genetic dysregulation, post-translational modifications, and altered cell signaling, promoting ligand-independent activation of ER and decreased sensitivity to antiestrogens [77,78,79]. The main resistance mechanism hinges on mutations affecting Estrogen receptor 1, frequently occurring after treatment with aromatase inhibitors [80,81,82], although the clinical implications of the response to fulvestrant, a third-generation aromatase inhibitor, remain unclear [78,81,83].

Other studies identified Phosphoinositide 3-kinase (PI3K) pathway dysregulation in metastatic BC progression and endocrine resistance [84,85]. *PIK3CA*, which encodes phosphatidylinositol-4, 5-bisphosphate 3-kinase catalytic subunit α, is mutated in approximately 40% of ER^+^/HER2^−^ advanced BCs, resulting in a gain-of-function phenotype and increasing downstream signaling and oncogenesis [84,86]. Most of the mutations occur at E542K or E545K in exon 9 (helical domain of p110α) and H1047R/L in exon 20 (activation loop in the kinase domain), representing “hotspot” mutations. The activating mutations in *PIK3CA* can cause hormones and targeted treatment resistance [84].

Concerning the HER2 pathway, around 25% of initial invasive breast tumors are characterized by the overexpression or amplification of *HER2*, correlating with aggressive cancer behavior and resistance to targeted therapies [87]. It is involved in various cellular processes, including cell proliferation, differentiation, migration, and invasion [88]. Despite therapies such as trastuzumab, a monoclonal antibody employed to stop cell growth, which significantly improves patient outcomes, the hurdle of drug resistance persists [89,90]. This can emerge through various mechanisms, such as the existence or onset of clones with different traits within the tumor, enhancing the adaptability in response to treatment [91].

CDK4/6 functions are crucial in the regulation of G1-to-S cell cycle phase transitions and are positively regulated through association with D cyclins (D1, D2, D3) and negatively by tumor suppressors, such as p16INK4A encoded by Cyclin Dependent Kinase Inhibitor 2A (*CDKN2A*), which prevents CDK4/6 from interacting with D-type cyclins [81]. Recently, three CDK4/6 inhibitors received Food and Drug Administration approval for ER^+^/HER2^−^ tumors treatment, frequently occurring in postmenopausal women. The inhibition of these kinases has proven strong preclinical efficacy in ER^+^ cases, particularly when paired with antiestrogen treatments [92]. However, intrinsic or acquired resistance can lead to disease progression in many patients, making it urgent to understand the involved resistance mechanisms [84,93].

### 4.2. Synthetic Lethality: PARPi for BRCA-Mutated Tumors

Standard treatment approaches, as well as first-line handling choices for BC, depend on tumor characteristics, biology, and behaviors [90]. However, innovative compounds have been devised to exploit the vulnerability of cancer cells with *BRCA* mutations, specifically targeting poly ADP-ribose polymerases (PARPs), indicating a viable approach. PARPs are the key components of the single strand breaks repair machinery. Mutations in *BRCA1* and *BRCA2* disrupt the normal functioning of the non-homologous end joining (NHEJ) DNA repair mechanism within cells. This machinery links the broken DNA ends, independently from a homologous template, acting as an error-prone mechanism and causing genomic alteration accumulation [94,95]. *BRCA*-mutated tumors generally have a high mutational burden, prompting researchers to discover that PARP inhibition causes the collapse of the replication fork, pushing the HR machinery to resolve the DNA breaks.

The PARPi mechanism of action involves PARP-1 protein binding at a single-stranded break or DNA lesion, interrupting its catalytic cycle, resulting in replication fork advancement, inducing double-strand breaks [96]. The ability of PARPi to bind PARP-1 NAD^+^ binding site provokes conformational changes, leading to the stabilization of interaction between PARP-1 and DNA, preventing the catalytic site from reverting to the inactive state. In this manner, PARP-1 lacks its standard function [97]. The accumulation of SSBs causes a stop of replication fork advancement. As a result, employing PARPi in BRCA-deficient cells selectively induces apoptosis without damaging the non-deficient cells. This condition is called “synthetic lethality” and takes place when the deletion of one gene is compatible with cell survival, but if another gene is disrupted concurrently, the cell dies [98,99]. To further enhance the efficacy of PARPi as anticancer agents, they are used in combination with radiotherapies or cytotoxic drugs, such as cisplatin [95], temozolomide, an alkylating agent [100,101], or camptothecin [102].

Although the initial positive impact of these therapies, patients commonly develop resistance. In the literature, various mechanisms of resistance to PARPi have been described. One consists of restoring BRCA1/2 functions by secondary mutational events occurring in *BRCA* genes, enabling the production of functional proteins, or by genetic reversion of the inherited loss of function mutations [103,104]. For instance, the reversion of *BRCA* mutations can remove frameshift mutation and restore the open reading frame [105]. This mechanism is clinically relevant after treatment with platinum. Gene reversion mutations can restore HR genes, both *BRCA1/2* and *RAD51C/D*, and overcome the effect of the PARPi, such as a partially functional NHEJ mechanism [106]. Another mechanism involved the loss of PAR glycohydrolase with the increase in the PAR chains. Additionally, resistance could depend on the inhibition of the degradation of the DNA replication fork performed by some nucleases [106]. In BRCA1/BRCA1-deficient cells, the levels of the Enhancer of Zeste 2 Polycomb Repressive Complex 2 subunit (EZH2) and Pax transcription activation domain-interacting protein activity are decreased, affecting the recruitment of nucleases and protecting the fork. Indeed, the PARPi cytotoxic effect is countered by an upregulation of the ATR/Chk1 pathway, which results in enhanced replication fork stability. Furthermore, the loss of PARPi-sensitivity could be related to the presence of wild-type Phosphatase and tensin homolog, modifying cell growth and cell proliferation [107].

As described by Giudice et al., a murine model revealed another mechanism of PARPi resistance [108]. An overexpression of drug-efflux transporter genes (Abcb1a and Abcb1b) has been observed in BRCA1-deficient breast tumors, in which increased drug-efflux of PARPi, removing them from cells and, consequently, reducing their effects [108].

Nonetheless, to overcome the resistance to PARPi, there is a need to fill the knowledge gap about BCs caused by *BRCA* mutations, as well as create new selection markers and novel molecules to target.

### 4.3. BRCA2 Deficiency, Genome Stability, and Sensitivity to Phytoestrogens and Radiation

BRCA2’s role as a transcription factor may denote a ‘caretaker’ and a ‘gatekeeper’ function by aiding in DNA repair [109]. BRCA2 is a tumor suppressor that exhibits recessive behavior, and the wild type allele is typically lost by tumor cells during the loss of heterozygosity process [110,111]. Most *BRCA2* mutations are nonsense, resulting in premature termination of translation, inducing the synthesis of a shorter C-terminally truncated protein.

Studies conducted on murine models have shown that the similarity between mouse and human nucleotide sequences of the *BRCA2* is 74% and 59% for protein [112]. Homozygous *BRCA2*^−/−^ mice typically die at the end of embryonic development or shortly after birth, with strain-specific differences [113]. *BRCA2*^−/−^ mice are smaller than wild type and exhibit spermatogenesis abnormalities. They also are more likely to develop malignant lymphomas, while heterozygous *BRCA2*^+/−^ animals are not tumor-prone [114]. Homozygous *BRCA2*^−/−^ mice are more vulnerable to ionizing radiation than their wild type counterparts, in agreement with the BRCA2 role in DNA repair [115]. Furthermore, *BRCA2*^−/−^ mice had significantly greater rates of DNA DSBs compared to *BRCA2*^+/−^ mice. DNA repair deficiency results in huge chromosomal changes involving translocations and deletions, a chromosomal phenotype missing in heterozygotes [116]. From this point of view, several researchers investigated the molecular consequences of transient or stable BRCA2 deficiency in human breast cell lines. Tripathi et al. [117] reported observations on the effect of *BRCA2* transient depletion in normal Human Mammary Epithelial Cells and BT549, derived from ductal carcinoma. Also, Bernard-Gallon et al. conducted a microarray analysis of mRNAs isolated from *BRCA2* knock-down cells showing 14 significantly downregulated genes [118]. Moreover, the downregulation of *BRCA2* in human breast cells with different phenotypes by RNA interference microarray revealed 35 differentially expressed genes by comparing MCF-7 (ER^+^/β^−^ or ER^−^/β^−^) and MDA-MB-231 (ER^−^/β^+^), using MCF-10a (ER^−^/β^−^) as reference. After exposure to phytoestrogens (genistein or daidzein), *BRCA1* was increased in knocked-down MCF-7 cells with *BRCA2*-siRNA, while *BRCA2* was increased in knocked-down MDA-MB 231 with *BRCA2*-siRNA. In the MCF-10a cell line, daidzein displayed a significant efficacy, decreasing the apoptosis-related genes *BAX* and *BCL2*, while both genistein and daidzein increased the tumor suppressor BRCA1 interacting DNA Helicase protein (*BRIP*) in *BRCA2*-silenced MCF-7 and MDA-MB-231 cell lines [118]. Upregulation of BRIP1 regulated by phytoestrogen, under *BRCA2* knock-down conditions, showed a possible connection between BRCA2, BRIP1, and phytoestrogen pathways. Another consequence of treatments with phytoestrogen and *BRCA2* knock-down conditions was the downregulation of *BCL2* and *BAX* gene expressions. These results underline different responses to phytoestrogens on BC dependent on ER status, suggesting that they may play a role in limiting genomic instability in cells lacking on *BRCA2,* by enhancing the expression of both *BRCA1* and *BRCA2* [118]. These findings sustain the link between apoptosis, *BRCA*-mutated BC, and phytoestrogen pathways. In fact, *BRIP1* could possibly be associated with BC susceptibility because of its relationship with BRCA1, directly interacting with its C-terminal domains. As validation of what has been said, BRCA1/2-associated BRIP1 activation is necessary for G1/S progression and this progression is delayed when BRIP1 and BRCA1 are depleted by silencing [118]. Further research can shed light on how BRIP1 operates in *BRCA*-mutated BCs [119]. A study by Castro et al. reconstructed the evolutionary scenario that linked apoptosis with genome stability pathways in a functional human gene/protein association network. They found that genome stability and apoptosis were engaged during evolution, recruiting genes that merged both systems. The apoptosis-related genes *BAX* and *BCL2* were discovered to be equally relevant to cancer as *BRCA1*, *BRCA2*, and *TP53* [120]. The interaction between these genes and *BRCA2* is summarized in Figure 2.

Taken together, these results suggested a potential chemopreventive effect of phytoestrogens in promoting apoptosis and maintenance of genome stability [121].

## 5. Single Cell Analysis: *BRCA1* and *TP53* Expression in *BRCA1*-Mutated Models

In this manuscript, we further focused on single cell RNA sequencing (scRNA-seq) experiments performed on *BRCA1*-mutated models which provide a high-resolution view of cellular diversity, enabling a thorough examination and validation of the molecular and functional findings. We identified two research articles not captured in the previous query due to absence of treatment. Nevertheless, we believe their inclusion offers a comprehensive evaluation derived from mutated models, which could be valuable for future investigations.

### 5.1. BRCA1 Mutation and the Cellular Origins of BC: Insights from Luminal Progenitors

Hu et al. tried to determine whether luminal cells are a source of the tumors due to impaired differentiation processes in luminal lineages by comparing their increase in normal breast tissues from *BRCA1*-mutated carriers [122]. The correlation analyses suggested that ER^−^ basal-like might originate from luminal progenitor cells, while ER^+^ luminal BCs might originate from mature luminal cells.

Noteworthy, the decrease in BRCA1/*TP53* expression and the increase in basal-mesenchymal transition in luminal progenitor cells from normal tissues might contribute to basal-like BC development in *BRCA1* carriers.

Data from scRNA-seq combined with bulk RNA-sequencing and immunohistochemistry (IHC) showed an increase in both luminal and basal progenitors derived from *BRCA1* carriers compared with noncarriers. The downregulation of Wnt, Notch, and Hedgehog signaling pathways in luminal cells might be responsible for the elevated number of luminal progenitors in *BRCA1* mutation carriers, preventing their differentiation with consequent accumulation. This is consistent with previous research indicating that Notch mechanisms are upregulated by BRCA1 [123].

Several studies have suggested an origin for the luminal progenitors of *BRCA1*-mutated/ER^−^ basal-like cancers [124,125], but the origin of those ER^+^ has not been explored. For this reason, data obtained from scRNA-seq confirmed and expanded earlier discoveries, showing a cell development trajectory based on single cell expression profiles. This analysis elucidates the cellular origins of various tumors, demonstrating a correlation between ER^−^ tumors and luminal progenitor cells, and between ER^+^ tumors and mature luminal cells. However, Hu et al. do not eliminate the possibility that ER^+^ epithelial cell expression profiles may vary throughout carcinogenesis, supporting the recent hypothesis that BRCA1-related basal-like BCs originate from luminal progenitors [126].

Of note, the luminal progenitors in normal breast tissues from *BRCA1* mutation carriers showed considerably lower amounts of *BRCA1* and *TP53* transcripts than in non-carriers, indicating that this highly proliferative cell population may require an adequate level of BRCA1 and p53 protein to preserve genome integrity during DNA replication.

Heterozygous germline mutations reduce *BRCA1* transcript levels in luminal progenitors of mutation carriers, leading to DNA damage accumulation in the normal portion, triggering p53-dependent checkpoint control and apoptosis [127]. Since *TP53* expression in these cells is lower than in noncarriers, the p53-dependent checkpoint control is potentially compromised in agreement with a higher proliferation of luminal progenitors [122,124,127].

Concerning the induction of the basal-mesenchymal transition in the luminal progenitors, scRNA-seq carried out in a mouse model showed an upregulation of transition features in these cells derived from *BRCA1* mutation carriers [128]. This suggests possible *BRCA1* haploinsufficiency and the onset of the basal-mesenchymal trait before tumorigenesis, which resulted in further enhanced during tumorigenesis. Predisposition to basal-like tumors in *BRCA1* mutation carriers can be explained by an increase in luminal progenitors, their long lifespan and proliferation capacity [129], and *BRCA1* and *TP53* transcript downregulation that creates the conditions for accumulation of mutations and luminal to basal-mesenchymal transition [59].

In conclusion, scRNA-seq data supports the cellular origin and evolution of BC in *BRCA1* mutation carriers, suggesting that aberrant luminal progenitors may contribute to basal-like phenotype by upregulating these characteristics before and during tumorigenesis, whereas ER^high^ luminal tumors could arise from mature luminal cells.

Taken together, these findings could help to adopt preventative measures in the cure for *BRCA1* mutation carriers [122].

### 5.2. Aberrant Alveolar Differentiation in Luminal Progenitors Drives Early Tumorigenesis in BRCA1/p53 Models

The second study aimed to clarify the effects of the mutations occurring in luminal progenitor cells in the presence of BRCA1’s loss of function. Bach et al. performed a time-resolved single cell profiling of genetically engineered mouse models, before tumor formation, demonstrating how BRCA1/p53 mutations in luminal progenitors are capable of aberrant alveolar differentiation during premalignancy; furthermore, the immune compartment is affected by pro-tumorigenic changes [130]. ScRNA-seq on cells isolated from mammary glands of 15 BRCA1/p53 mice, grouped in various premalignant stages and fully developed tumors, allowed us to identify the changes in the abundance of the 16 cell types during the early stages of tumorigenesis. Among them, the only epithelial cluster expanding was a luminal cell cluster with a secretory alveolar profile. This was virtually absent at Stage 1 and formed more than a third of the epithelium, appearing as the most proliferative of the entire tissue, while in homeostasis conditions, these cells were restricted to gestational and lactational stages [131] and derived from hormone-mediated differentiation of luminal progenitors (LP) [132]. Differentiation of LPs towards the alveolar fate always occurred, associated with the expression of known markers of alveologenesis, such as the milk protein beta-casein (CSN2) and the transcription factor E74-like ETS transcription factor 5 (Elf5), despite all animals being nulliparous. The presence of alveolar cells was confirmed by using immunofluorescence.

Finally, the authors performed a transposase-accessible chromatin sequencing (ATAC-Seq) in LPs of BRCA1/p53 animals before tumor onset. Data showed increased accessibility at several key genes of alveologenesis, such as *CSN2* and Whey Acidic Protein, whose proximal enhancer regions are more accessible during gestation [130]. These regions showed significant enrichment for key transcription factors involved in alveolar differentiation, including CCAAT enhancer binding protein *β* (CEBPB), Elf5, Nuclear Factor κ-B subunit 1 (NF-κB1), and SRY-box transcription factor 10, suggesting that, in early phases of carcinogenesis in the BRCA1/p53 mouse model, the luminal progenitors can turn gradually toward the alveolar fate [130].

The expression of *CSN2* was determined by using qPCR in FACS-sorted luminal progenitor cells from *BRCA1* mutation carriers who had undergone a prophylactic mastectomy and healthy women undergoing a reduction mammoplasty to obtain any information of what might happen in human BC development. Two samples from *BRCA1* mutation carriers were distinguished for elevated *CSN2* levels. To further validate the data, RNA-seq analysis was carried out on independent samples obtained from luminal progenitors of healthy subjects and *BRCA1* mutation carriers, demonstrating the latter high levels of *CSN2*, with the enrichment of pathways involved in the recruitment of the immune system as well as positive regulation of NF-κB. Although a study carried out on human samples may not give information on the cellular and temporal evolution compared to the mouse model, overall, these findings suggest that aberrant differentiation of luminal progenitors occurred also in humans.

To further characterize the aberrant alveologenesis, they compared it to the homoeostatic counterpart at three gestational time points, 4.5, 9.5, and 14.5 day of gestation (dG), and integrated it with the tumorigenesis data. Systemic hormones, such as progesterone produced by the corpus luteum, drive epithelial development throughout pregnancy. As a result, transcriptional responses have been observed in every epithelial compartment. It is well known that hormone-sensing cells directly react to pregnancy hormones, releasing paracrine signaling factors like TNF superfamily member 11 (TNFSF11, also called RANKL) and insulin-like growth factor 2 to control tissue development. In this manner, the basal compartment increases the production of many collagens and myosins necessary for the ducts to contract when the baby feeds. Researchers observed the gradual differentiation of luminal progenitors, which started at 4.5 dG, reaching near completion at 14.5 dG. This event was marked by the expression of various milk proteins and genes involved in fatty acid metabolism.

In contrast, in the aberrant phenotype of the BRCA1/p53 animals, hormone-sensing luminal cells lacked the transcriptional response observed during pregnancy, dependent on the absence of progesterone signaling. The lack of basal differentiation during BRCA1-mediated carcinogenesis supports this hypothesis, showing that the cell-autonomous process is limited to the luminal progenitor compartment and the gestation-like phenotype is hormone-independent.

This analysis exhibited 137 genes that differed in their linkage to *CSN2*. During tumorigenesis, no correlation emerged between *CSN2* and numerous genes involved in fatty acid metabolism, which normally increases during gestation. This implies that the alveolar cells found during early stages do not fully function as secretory cells. During tumorigenesis, genes with a positive correlation to *CSN2* included many factors associated with basal-like BC, such as CEBPB. Interestingly, multiple studies indicated NF-κB, among the regulators of alveologenesis, especially induced in response to DNA damage [125,130,133]. The abundance of CEBPB and NF-κB binding sites in the exposed chromatin of luminal progenitors suggests that this reaction may inadvertently trigger a transcriptional program of alveolar differentiation.

This implies that early stages of cancer development in BRCA1/p53 models are mainly characterized by events involving the cell-autonomous differentiation of progenitors in the luminal compartment and, in contrast to normal development, aberrant differentiation precedes *RANKL* expression in hormone-sensing cells.

## 6. Novel Therapies Suggested in the Management of *BRCA1*-Mutated Tumors

Several studies have proven the enhanced risk for women who carry the *BRCA* mutations to develop secondary cancer. In this case, the recommendation is a bilateral mastectomy to reduce the risk of dying of BC. Despite being often negative for hormones, they can still be treated with chemotherapeutic, more effective if compared with sporadic BC. Indeed, the BRCA-mutated BC exhibits specific clinical characteristics, such as increased sensitivity to PARPi, DNA-damaging agents such as platinum-based chemotherapies, and potentially decreased sensitivity to CDK4/6 inhibitors. Prompt determination of BRCA status is crucial to establish effective treatment strategies for patients.

Targeted drugs, immunotherapies, and combined treatments are being developed to maximize effectiveness and minimize side effects, providing new hope for *BRCA*-mutated patients. Research on novel therapies for malignancies linked to *BRCA* mutations is a fast-moving field that aims to improve patient prognosis and quality of life. Individuals carrying *BRCA1/2* mutations face an increased risk of developing breast, ovarian, prostate, and pancreatic cancers. Consequently, therapeutic strategies that exploit the specific abnormalities of cancer cells are also applicable to hereditary breast cancer and allow for the utilization of PARPi, as previously mentioned [2].

Despite the potential of platinum-based chemotherapies, PARPi, and CDK4/6 inhibitors, a large number of patients develop resistance to treatment or toxicity, which represents the most important problem [2]. This subtype of BC remains a big challenge. Advancements in this field are progressing rapidly with several potential new avenues emerging. Although PARPi potential has been largely explored, the resistance to this treatment is a big issue. Currently, the combinations of conventionally used BC therapies together with new molecules are under investigation to improve patient outcomes. Similar to PARPi treatments, synthetic lethality is being employed by other molecules, such as BETi [134].

Our analysis evidenced relevant manuscripts which allow us to create interconnections focused on the deregulated genes by *BRCA* mutations and further modulated upon treatments [50,118,134,135,136,137,138,139,140,141,142,143,144,145]. In this context, experiments carried out in cell cultures or animal models are schematized in Figure 3 and further explored in the following paragraphs.

These molecules can represent interesting therapeutic tools to fight this specific cancer subtype. Data reporting IC50, dosage, and efficacy (when indicated) were summarized in Table 1.

Concerning PARP, although PARPi monotherapy produces a minimal impact on the overall survival of patients, they can enhance immunotherapy effectiveness.

KEYNOTE-162 (niraparib + pembrolizumab) and MEDIOLA (olaparib + durvalumab) trials explore PARPi-immunotherapy combinations, and the JAVELIN (avelumab + talazoparib) trial demonstrated an acceptable safety profile [16]. However, further investigation is necessary.

### 6.1. Efficacy of BET Inhibitors and Their Effects on BRCA1 Deficient Cells

Zhang et al. [134] demonstrated the synthetic lethality interaction between BRCA1 and the bromodomain and extra-terminal domain (BET) to evaluate novel effective approaches contrasting the onset of drug-resistance. BET proteins are characterized by bromodomains (BRDs) with conserved BD1 and BD2 sequences at the N-terminals and an extra-terminal structure at the C-terminal. Their capacity to identify acetylated residues on histones H3 and H4 is enhanced by the presence of multiple acetylated fragments consisting of 1–5 amino acids. Among the BET family members, BRD4, also known as the lysine acetylation reader, is the most extensively investigated. It acts as a transcription factor regulator, specifically binding to acetylated histone tails, and recruiting tumor-associated target genes, modulating promoters and/or enhancer regions. BRD4 is essential for transcriptional regulation and controls the expression of many enhancer-associated genes, including *Myc.* This points out BETs as a promising target [148].

Transcriptome study of cells treated with BET inhibitors (BETi) indicated a significant shift in the expression of genes related to oxidative stress, including thioredoxin interacting protein (TXNIP). This protein forms a disulfide bond with TXN, reducing and inhibiting its antioxidant properties. It also acts as a tumor suppressor, affecting cell metabolism and cell cycle progression. In BC, TXNIP expression is closely linked to tumor progression and treatment outcome. As observed in the *BRCA1* isogenic TNBC cells, the basal levels of ROS and oxidative stress-response genes were higher in BRCA1-deficient cells than in BRCA1-wild type cells. Consequently, BETi evoked much higher oxidative stress responses in BRCA1-deficient cells, leading to severe DNA strand breaks and driving the cell to apoptosis. However, these effects can be reversed by using antioxidant treatment [134].

In drug sensitivity tests on a panel of BC cells, BETi showed better selectivity in the treatment of *BRCA1* mutants than wild type cells. *BRCA1*-mutated/BETi-sensitive HCC1937 and SUM149 cell lines showed a decreased *Myc* expression and increased *TXNIP* levels upon BETi treatment, unlike MDA-MB-436 cells, which are BRCA1-deficient/BETi-resistant. The silencing of BET with siRNA successfully reduced *Myc* and increased *TXNIP* levels in MDA-MB-436 cells, inhibiting cell viability and restoring the BETi response. BRCA1-wild type T47D cells were resistant both to BETi and siRNA treatments, although *Myc* and *TXNIP* decreased after BET inhibition.

These findings imply that there is synthetic lethality between BRCA1 and BET, but BETi probably cause resistance in certain *BRCA1* mutant cells. In these mechanisms, the Myc-TXNIP axis-mediated oxidative stress and DNA damage play a significant role. Finally, the pharmacological blocking of BET proteins may be a viable strategy to cure TNBCs with *BRCA1* loss-of-function mutations [134]. Indeed, the combined therapy of talazoparib, a PARPi useful to affect *BRCA1/2*-mutated HER2^−^ BC [149], and BETi is under evaluation in Phase II clinical trials to reduce solid tumors that are difficult to treat and could easily metastasize [150].

### 6.2. CDDO Treatment Impacts Survivin Expression

Our investigation evidences another interesting gene, Baculoviral IAP repeat containing 5 (BIRC5), also called survivin, implicated in the inhibition of apoptosis and in the control of cell division. Its overexpression has been reported in many types of tumors, suggesting a role in tumor growth viability maintenance of cancer cells, further supported by the influence of apoptosis induction after the inhibition of the gene [151].

Kim et al. [137] explained the effects of the synthetic triterpenoid 1-[2-cyano-3,12-dioxooleana-1,9-dien-28-oyl] imidazole (CDDO-Imidazolide, CDDO-Im) as a promising anticancer and chemopreventive agent. Experiments revealed that CDDO-Im treatment of non-malignant breast epithelial cells and *BRCA1* wild type cell lines does not affect ROS generation, demonstrating unique impact on *BRCA1*-mutated BC cells, in which CDDO-Im stimulates ROS and, consequently, DNA damage. This in turn helps the *BRCA1*-mutant cancer cells to activate the DNA damage checkpoint, enter G2/M arrest, and ultimately undergo apoptosis. The combination of CDDO-Im and uric acid acts against ROS generation, avoiding the DNA damage effect caused by CDDO-Im treatment. A proteome profiling carried out on *BRCA1*-mutated W780 and W0069 cells evidenced that CDDO-Im significantly increases caspase 3 cleavage, increases heat shock protein 70 (HSP70), and decreases in claspin, BIRC5, B-cell Lymphoma-extra-large, and BCL2 associated agonist of cell death. CDDO-Im, especially in W780 cells, caused the cleavage of PARP [137].

### 6.3. Resveratrol Modulates BIRC5 and SIRT1 in BRCA1-Mutated Models

Wang et al. [138] further explored the relationship between *BRCA1* mutation, BIRC5, and sirtuin 1 (SIRT1), its downstream effectors. SIRT1, a mammalian counterpart of yeast SIR2, is a NAD^+^-dependent type III histone and a protein deacetylase [152,153], which deacetylates various non-histone proteins, including p53 and the translational elongation factor II. The involvement of SIRT1 is crucial in several biological processes, such as apoptosis, neuronal protection, calories restriction adaptation, organ metabolism and functions, cellular senescence, and aging [154,155].

However, the role of SIRT1 in tumor development remains unclear. BRCA1 deficiency reduces SIRT1 and enhances BIRC5 levels. In untreated cells from mammary tumors lacking full-length BRCA1, SIRT1 was significantly lower compared to the *BRCA1*-wild type, as well as in transgenic mice models. SIRT1 reduction was also observed in two human *BRCA1*-mutated cell lines, HCC1937 and L56Br-C1, when compared to sporadic BC cells BT20 and MCF-7.

To investigate the SIRT1 impact on BIRC5 expression, researchers introduced SIRT1 vector into SIRT1^−/−^ mouse embryonic fibroblasts to restore its expression, obtaining decreased BIRC5 levels. A luciferase reporter assay [155,156] was then performed to analyze the activity of BIRC5 promoter following SIRT1 transfection. The results showed that SIRT1 overexpression decreased BIRC5 by approximately four-fold. Further analysis demonstrated that resveratrol also increased SIRT1 activity, leading to reduced BIRC5 expression, and acted as an inhibitor in *BRCA1*-mutants in vitro and in vivo. In fact, SIRT1 negatively regulates BIRC5 expression through its deacetylase activity, resulting in the epigenetic modification of the BIRC5 promoter and turning the promoter into a transcription-silent configuration. Besides that, resveratrol is a potent inhibitor for the initiation and progression of BRCA1 mutant cancer both in vitro and in vivo, making it an excellent molecule for targeted therapy [156].

### 6.4. EZH2 Differential Expression in BRCA1-Mutated BC After DZNep Exposure

BC carrying *BRCA1* mutations develops early, exhibiting a basal-like phenotype, and is linked to poor survival. Puppe et al. [144] studied the EZH2, a polycomb group protein responsible for the repression of cell cycle inhibitors and genes involved in the differentiation [157,158,159]. Since its overexpression has been described in BRCA1-deficient mouse mammary tumors [160,161], researchers investigated whether EZH2 was essential for cancer cell survival or a tumorigenic byproduct, demonstrating an EZH2-dependence, while BRCA1-intact cells tolerate EZH2 loss. EZH2 may influence DNA repair in BRCA1-deficient cells by repressing *Rad51* family genes [162] and affecting DNA damage signaling. On the other hand, *BRCA2*-mutation carriers did not show an increase in EZH2 expression, suggesting EZH2 oncogenic role is not tied to DNA repair.

The selective EZH2 overexpression in BRCA1-deficient tumors might depend on the cell of origin. IHC carried out in *BRCA1*-mutated cells revealed higher EZH2 levels in basal-like than in luminal A BCs [163,164,165], pointing out EZH2 as a therapeutic target. Additionally, BRCA1 absence is linked to stem cell traits and prevents luminal differentiation [166,167]. Hence, EZH2 is crucial for stem cell maintenance, resulting in high expression in undifferentiated BCs, in which it silences lineage-specific differentiation transcription factors [168,169].

Tan et al. found that 3-deanzaneplanocin A (DZNep) decreased H3K27 trimethylation (H3K27me3) and transcription of Polycomb Repressive Complex 2 (PRC2) partners, such as EZH2, Suppressor of zeste 12 (SUZ12), and embryonic ectoderm development [170]. This agrees with EZH2-H3K27me3 activity [171]. DZNep enabled them, promoting selective mortality in BRCA1-deficient BCs and triggering differentiation. Conversely, the inhibition of H3K27me3 in numerous cancer cell lines caused apoptosis and reactivated PRC2-silenced genes. To demonstrate if growth in BRCA1-deficient cells was mediated by EZH2 inhibition, both DZNep and siRNAs against EZH2 were employed, demonstrating a more efficient decrease with DZNep than with specific siRNA, suggesting the interference also of other epigenetic markers [170,171].

Since BRCA1 loss induces genomic instability, the EZH2 reliance was investigated to define whether it was associated with BRCA1 loss rather than tumorigenic mutations. Reintroducing a BAC clone containing the full-length BRCA1 into a BRCA1-deficient cell line, clones expressing *BRCA1* with low cisplatin sensitivity, demonstrates that BRCA1 is functional, while EZH2 levels were unaltered in these reconstituted cell lines, suggesting that BRCA1 does not directly affect EZH2 expression [171].

## 7. Pathway Involving Deregulated Genes by BRCA1

We examined the interactions between genes in the perspective to define various pathways based on the results of the query employed as methods of article selection, as shown in Figure 4. We used the list of genes involved in different processes, such as DNA replication and cell cycle, or biological pathways like metabolism and apoptosis, already used to create the networks of Figure 2 and Figure 3, also reported in Appendix A. For example, we found that BRCA1 interacts with and influences the expression of *SIRT1*, *BIRC5*, and *EZH2* genes, which are involved in chromatin remodeling, apoptosis inhibition, and stem cell maintenance. Two interesting pathways emerged from our analysis, proteoglycan pathways and ferroptosis. On one side, proteoglycans are a structural part of the malignant cells forming the tumor, and molecular changes in these molecules could impact cell properties. On the other side, ferroptosis is particularly important in tumor suppression because it drives cells to death. Moreover, the introduction of this process has been associated with the reversion of drug resistance [172]. We analyzed further these pathways in the next sections.

### 7.1. Proteoglycan Pathway

Proteoglycans were identified in cancer biology over 50 years ago when pathologists observed that some carcinomas induced a desmoplastic hyperproliferative reaction in the host stroma and surrounding connective tissue [173]. Advances in molecular and cellular biology have also clarified their specific roles in cancer. Proteoglycans play an important function in cellular settings, contributing to cancer and angiogenesis processes by interacting with ligands and receptors that control tumor growth and blood vessel formation. However, some proteoglycans, such as perlecan, can have both pro- or anti-angiogenic properties, whereas others, like syndecans and glypicans, affect relevant signaling pathways, inhibiting tumor growth.

In the proteoglycan pathway, decorin significantly decreases Myc, among the targets of β-catenin, promoting its phosphorylation and causing instability and subsequent proteasomal breakdown [174,175]. Under proliferative conditions, Wnt signaling and hyperactive RTK pathways stabilize β-catenin and Myc via GSK-3β phosphorylation, releasing active forms of these proteins. These activated proteins then move to the nucleus, triggering numerous genes, including *Myc* itself, establishing a positive feedback loop that supports growth and survival [176].

The *Myc* role in the regulation of cell proliferation or apoptosis is context- and isoform-dependent [177,178]. *Myc* amplifications are more common in tumors of *BRCA1* mutation carriers [179,180,181,182], but not all *BRCA1*-mutated tumors exhibit this, suggesting a possible connection between the type of *BRCA1* mutation and *Myc* amplification. Previous studies have identified two regions in BRCA1 that independently interact with Myc, involving amino acid residues 175–303 and 343–433. These regions span exons 8, 9, and 10 and the N-terminal portion of exon 11. *Myc* amplification was observed in tumors with *BRCA1* mutations located upstream or within Myc-binding sites, as well as in cases with mutations downstream of these sites. It seems that regions downstream of Myc-binding sites may indirectly affect the BRCA1-Myc interaction, or the resulting truncated BRCA1 protein might be unstable and unable to interact effectively with Myc.

*Decorin*, a prototype member of the small leucine-rich proteoglycans, is another gene involved in the proteoglycan pathway in cancer [183,184]. It is named for its ability to ‘decorate’ collagen fibers at specific locations and regulate fibrillogenesis both in vitro and in vivo [185,186,187,188]. Unlike some proteoglycans, certain small leucine-rich proteoglycans, such as decorin and lumican, act as tumor suppressors by physically antagonizing receptor tyrosine kinases like the EGFR and Met receptors or integrin receptors, activating anti-survival and pro-apoptotic pathways. Decorin binds to a specific region of the EGFR that partially overlaps but is distinct from the EGF-binding domain [189]. The interaction between decorin and EGFR results in sustained downregulation of the EGFR [189]. In mammary carcinoma cells, decorin also interacts with ErbB4 and indirectly reduces ErbB2 levels, likely by decreasing ErbB2/ErbB4 heterodimers [190].

From the therapeutic point of view, decorin expression strongly prevent processes associated with spread of metastasis, also targetable by drugs, such as stachydrine [191].

### 7.2. Ferroptosis Pathway

Ferroptosis, a unique form of cell death dependent on intracellular iron, differs from apoptosis, necrosis, and autophagy. Research indicates that ferroptosis is crucial for tumor suppression, offering new avenues for cancer treatment. Interestingly, ferroptosis has been linked to cancer therapy resistance, and its induction has been shown to counteract drug resistance [172].

The iron-dependent nature of ferroptosis is partly explained by the reliance of lipid peroxidation on both iron-dependent enzymes and iron-mediated Fenton reactions. Thus, altering iron metabolism can affect susceptibility to ferroptosis.

Heme, derived from hemoglobin and myoglobin, is the primary source of iron in mammals. Heme oxygenase 1 (HMOX1) breaks down the heme into carbon monoxide (CO), biliverdin, and free iron. Biliverdin reductase then converts biliverdin to bilirubin. Various stimuli, including cytokines, endotoxin, heat shock, and heavy metals, can activate HMOX1, suggesting its potential role in redox homeostasis maintenance.

HMOX1 exhibits a dual function in ferroptosis induction. In HT1080 fibrosarcoma cells, erastin, a small molecule compound, is capable of selectively killing tumor cells, which has been implicated in ferroptosis. In fact, erastin stimulates HMOX1 expression, potentially promoting ferroptosis. Hemin, an HMOX1 inducer, accelerates erastin-induced ferroptosis in these cells, while zinc protoporphyrin IX, an HMOX1 inhibitor, hinders erastin-induced ferroptotic cell death. Moreover, CO-releasing molecules have been found to enhance erastin-triggered ferroptotic cell death [192], implying that HMOX1-generated CO might act as an endogenous pro-ferroptosis molecule.

Conversely, HMOX1 can sometimes prevent ferroptosis. For example, erastin and RSL3 treatments increase HMOX1 expression in renal proximal tubular cells. *HMOX1* knock-down enhances erastin- or RSL3-induced cell death in liver cancer cells [193] or renal proximal tubular cells [194]. Thus, HMOX1 role during ferroptosis may be context-dependent [195].

Introducing holo-transferrin as an iron carrier protein also triggers ferroptotic cell death in amino acid-deprived *BAX/BAK* double-knockout cells [196].

Inhibiting *HMOX1* pharmacologically or through silencing confers resistance to ferroptosis induced by withaferin A, erastin, and BAY 11-7085 [197,198].

However, HMOX1 can also act protectively, likely depending on its activation level. The protective effect of HMOX1 is attributed to its antioxidant activity, while its toxic effect results from increased ferrous iron production, which enhances Fenton-mediated peroxide decomposition when ferritin buffering capacity is insufficient. Also, excessive *HMOX1* upregulation may be cytotoxic, while moderate upregulation could be cytoprotective [199,200]. Wu et al. [201] aimed to prove solute carrier family 3 member 2 (SLC3A2) expression in various cancers apart from the mechanism of this carrier in ferroptosis. SLC3A2 is a molecular chaperon of light chain subunits, capable of regulating amino acid transport, leading to an important role in tumor growth and oxidative stress control. Due to these functions, the overexpression of SLC3A2 and SLC7A11 is related to the occurrence and development of various types of cancer. When comparing BC with the normal counterpart, an increase in BC tissue was observed for SLC7A11 and SLC3A2 [202]. These ferroptosis carriers are also affected by the production of interferon γ (IFNγ) which leads to ferroptosis, linked to lipid peroxidation in tumor cells. IFNγ produced by CD8^+^ T cells triggers the JAK/STAT1 pathway, downregulating SLC7A11 and SLC3A2, which makes tumor cells more vulnerable to ferroptosis [203].

Concurrently, IFNγ can upregulate Acyl-CoA Synthetase Long Chain Family Member 4 expression through STAT1/IRF1 signaling, facilitating the integration of tumor microenvironment-associated arachidonic acid into phospholipids, inducing ferroptosis in tumor cells [204]. Another class of carriers involved in ferroptosis is solute carrier family 11 member 2 (SLC11A2), which allows the release of endocytosed ferrous iron into the cytoplasm, creating the labile iron pool, capable of catalyzing hydroxyl radical formation and initiating ferroptosis [205]. Excess intracellular iron is typically stored in ferritin, a protein made up of two subunits: ferritin heavy chain 1 and ferritin light chain [206]. The latter undergoes degradation through ferritinophagy, a process mediated by nuclear receptor coactivator 4, leading to the release of large amounts of iron [207,208]. Furthermore, an excess of cytoplasmic ferrous iron can be exported from the cell through solute carrier family 40 member 1 [209,210]. A promising immunotherapy approach involves ICIs, which target compromised immune systems and primarily activate CD8^+^ T cells to efficiently eliminate tumor cells [211]. The development of ICIs, especially anti-CTLA4 and anti-PD-1/PD-L1 antibodies, has transformed cancer treatment and marks a significant breakthrough in oncology [212]. Moreover, the effectiveness of anti-tumor immunotherapy is enhanced by ferroptosis, since CD8^+^ T cells simultaneously promote ferroptosis in tumor cells [213,214,215,216].

Transferrin binds to ferric iron, the predominant form of iron in the bloodstream [217], and ferric iron enters cells via TFR1 to finally localize in endosomes [218]. The metalloreductase encodes a multipass membrane protein that functions as an iron transporter. The encoded protein can reduce both iron (Fe^3+^) and copper (Cu^2+^) cations. This protein may mediate downstream responses to p53, including the induction of apoptosis [219,220].

The complex connection between iron intake, storage, and export pathways emphasizes how important iron homeostasis is to cellular function and how a disruption of these interactions could contribute to ferroptosis. Currently, the ferroptosis pathway is acquiring relevance for its impact on patients’ survival [221], ALB (included in our list) is especially reported as a prognostic factor [222], and the induction of ferroptosis by SLC7A11 targeting sensitizes *BRCA*-mutated and PARPi resistant cancer [223]. Ferroptosis induction could be an interesting strategy to overcome PARPi drug resistance in BC where this way could be activated, such as p53-positive tumors [220].

Gaining knowledge on these pathways might help researchers discover possible treatment targets for modulating cellular processes that depend on iron and prevent ferroptotic cell death.

## 8. Conclusions

This review explored the intricate relationship between *BRCA1/2* mutations, changes in gene expression, and the biological mechanisms underpinning BC development and its treatment. *BRCA1/2* mutations critically disrupt DNA repair mechanisms, genomic stability, and cell cycle regulation, fostering tumorigenesis through the accumulation of DNA damage. ScRNA-seq studies have offered valuable insights, identifying luminal progenitor cells as potential origins of basal-like BC in *BRCA1* mutation carriers, characterized by decreased BRCA1 and p53 expression. Therapeutic advancements targeting the unique vulnerabilities of *BRCA*-mutated tumors, such as PARPi, represent a considerable promise in both early and metastatic stages. However, drug resistance remains a significant obstacle. BET inhibitors (BETi) have emerged as promising agents exploiting synthetic lethality, particularly by triggering oxidative stress-related apoptosis through Myc and TXNIP production in *BRCA1*-deficient models. Moreover, DZNep treatment has selectively eliminated *BRCA1*-mutated cells by modifying EZH2 transcription, a crucial factor in DNA repair and cancer cell proliferation. In addition to PARPi, chemopreventive agents like phytoestrogens have been investigated for their capacity to induce apoptosis and maintain genome stability, resulting in increased *BRCA1/2* expression in *BRCA2* knockdown cells. Another potential agent, CDDO, an imidazolide, can stimulate ROS production and DNA damage, inducing apoptosis in *BRCA1*-mutants. These findings emphasize the importance of a tailored treatment strategy, combining targeted therapies with immunotherapies or metabolic interventions to improve clinical outcomes. Despite the fact that significant progress has been achieved, further research is essential to deepen the understanding of the molecular mechanisms involved and to develop innovative strategies that lead to more effective therapies for patients affected by *BRCA*-mutated BC. From this study, some interesting pathways suggest novel targetable processes. Ferroptosis has garnered attention as a novel therapeutic target. Its activation may overcome drug resistance and enhance the effectiveness of current treatments by targeting genes involved in oxidative stress and lipid peroxidation in those tumors in which it could be inducted. Also relevant, the modulation of proteoglycans in the extracellular space, where decorin could be induced, for instance, by stachydrine administration. We believe that the expanding role of precision medicine, including single-cell sequencing, will refine future therapeutic strategies and their applications of genetic testing to specific molecular subtyping.

## Figures and Tables

**Figure 1 biology-14-00253-f001:**
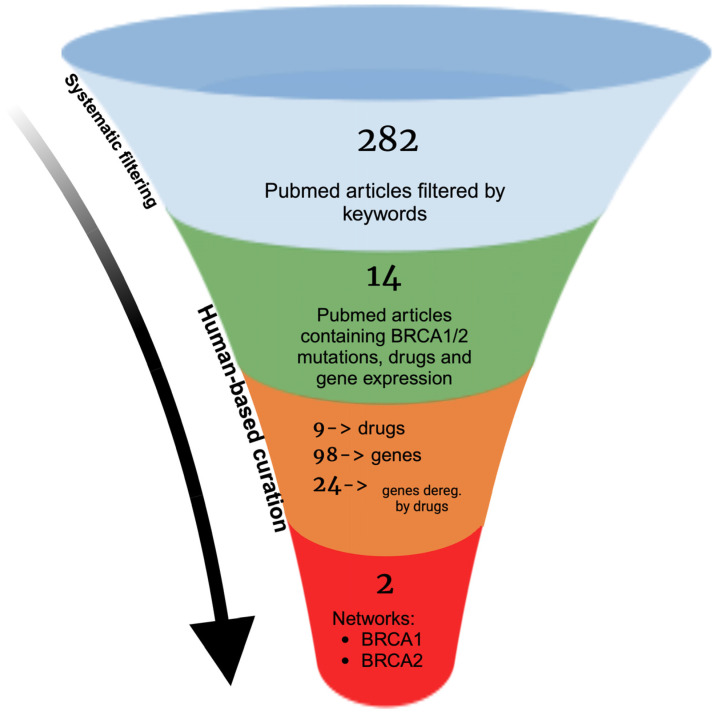
Scheme of progressive steps for the selection of articles including all criteria: *BRCA* mutations, modulated genes and drug treatments for network creation.

**Figure 2 biology-14-00253-f002:**
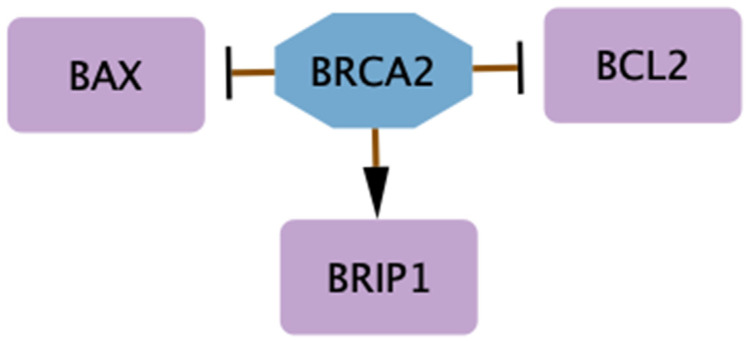
Link between apoptosis and genomic instability in BC cells. For *BRCA2*, we analyzed studies and evaluated the most impacted genes. In this case, *BRCA2* was represented as the “node” in the blue octagon, whilst the other genes were the “edges” in purple rectangles. Flat-tipped lines represent downregulation, whereas pointed arrows represent upregulation, and the brown lines are representative for phytoestrogen treatment.

**Figure 3 biology-14-00253-f003:**
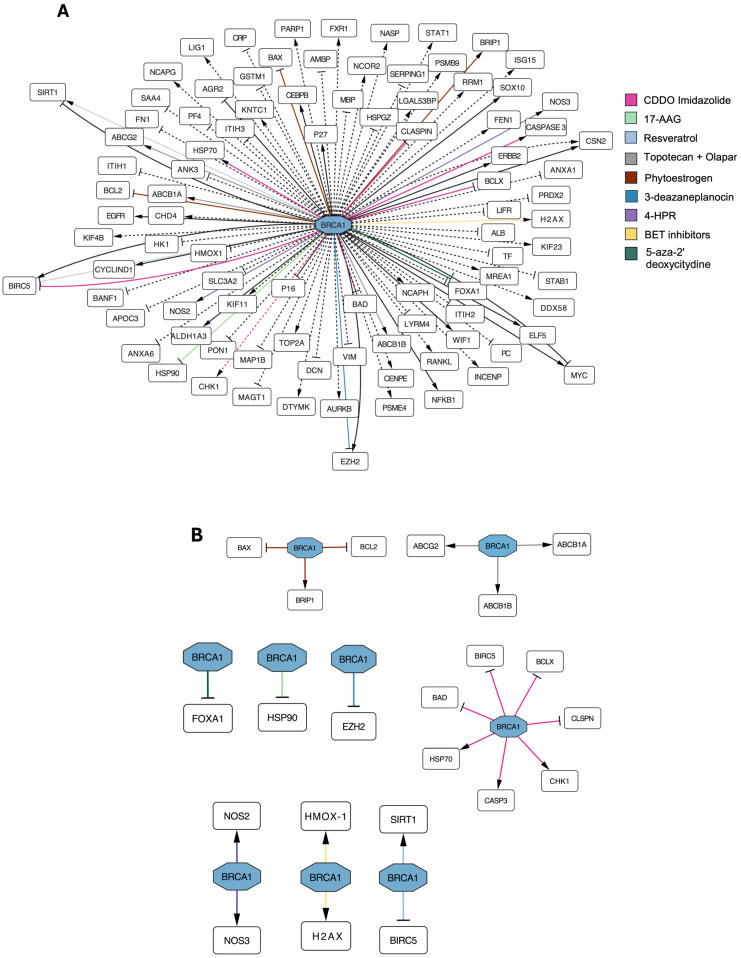
Network of correlation between altered genes by *BRCA1* mutation and treatments. (**A**) We used colors to graphically highlight *BRCA1* (light blue octagon) and the deregulated genes (white rectangles). *BRCA1* was represented as the “node”, whereas the other genes were named “edges”. Flat-tipped lines represent the downregulation of gene expression, while the pointed arrows represent upregulation. The colored lines represent the different treatments used on *BRCA1*-mutated cell lines and their effect on gene expression. Dashed lines are used to depict secondary interactions whereas the continuous lines portray primary and direct interactions. (**B**) Highlighting of the genes modulated by therapies.

**Figure 4 biology-14-00253-f004:**
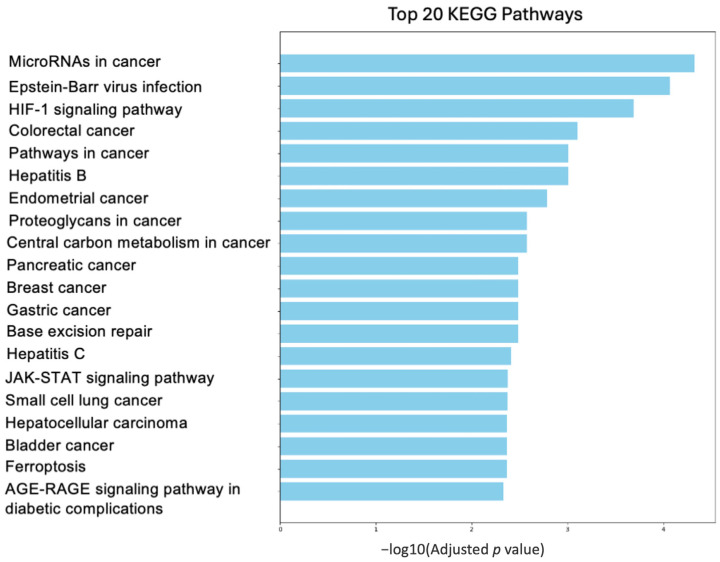
Bar chart of the top 20 KEGG pathways. The image was created with Python v.3.13.1 (matplotlib).

**Table 1 biology-14-00253-t001:** Drugs used in the articles considered to create the networks.

DRUGS	IC50 IN *BRCA*-MUTATED MODELS	DOSAGE	EFFICACY
topotecan + olaparib	topotecan 5.780934 μmol/L + olaparib 198.853952 μmol/L	n.d.	n.d.
17-AAG	0.059 ± 0.017 (μM) (MDA-MB-436) 0.014 ± 0.006 (μM) (HCC 1937) 0.013 ± 0.006 (μM) (UACC 3199)	n.d.	n.d.
CDDO-Im	n.d.	1 μM	n.d.
resveratrol	40 μM	n.d.	n.d.
DZNep	163 nM	n.d.	n.d.
4-HPR	n.d.	2.5 μM	n.d.
genestein	28 μM	n.d.	n.d.
daidzein	n.d.	1.7 mg/day	CI95% daidzein intake/recurrence: 0.96 (0.52–1.76) [146]
BETi	108.07 μM	n.d.	n.d.
5-aza 2′deoxycytidine	n.d.	0.1 μM 5-AZA-dC [147]	n.d.

## Data Availability

Data are available in the Appendix A of this article.

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
