# Peer review of "Druggable Molecular Networks in BRCA1/BRCA2-Mutated Breast Cancer"

_biology, 2025, doi:10.3390/biology14030253_

Round 1
Reviewer 1 Report
Comments and Suggestions for Authors
1. The abstract provides a concise overview but lacks quantitative details on key findings (e.g., the number of genes identified or specific drug responses). Including these would enhance clarity.
2. The explanation of triple-negative breast cancer (TNBC) subtypes could benefit from a more concise presentation. A few claims are not adequately supported by recent citations (e.g., TNBC classification based on Lehmann).
3. The inclusion criteria for article selection are overly broad and should be refined for reproducibility. Additionally, the use of only 15 articles out of 305 raises concerns about selection bias. The reference to WebGestalt should include the exact version and access date for reproducibility.
4. The results section occasionally lacks quantitative rigor, especially in describing drug efficacy and gene expression analysis. The pathway analysis is discussed but not linked clearly to clinical outcomes. Citations for pathway analysis should include primary sources from KEGG and related studies.
5. Some sections are repetitive, particularly concerning synthetic lethality and PARP inhibitors. There is limited discussion on resistance mechanisms and future research directions. Some broad claims about therapy efficacy lack specific references.
6. The conclusion summarizes the key findings effectively, but it could be more forward-looking by suggesting specific research gaps and clinical implications.
7. Several references are outdated, particularly in the immunotherapy section. Ensure the inclusion of recent studies (2020–2024). There are inconsistencies in formatting (e.g., missing DOIs and access dates).
Author Response
Comments and Suggestions for Authors
Reply to Reviewer 1
- The abstract provides a concise overview but lacks quantitative details on key findings (e.g., the number of genes identified or specific drug responses). Including these would enhance clarity.
Answer: We thank the Reviewer for pinpointing this omission in our abstract. We modified it as follows: “Mutations in the tumor suppressor genes BRCA1 and BRCA2 are associated with triple-negative breast cancer phenotype, particularly aggressive and hard-to-treat tumors lacking estrogen, progesterone, and human epidermal growth factor receptor 2. This research aimed to understand the metabolic and genetic links behind BRCA1 and BRCA2 mutations and investigate their relationship with effective therapies. Using Cytoscape software, two networks were generated through a bibliographic analysis of articles retrieved from the PubMed-NCBI database. We identified 98 genes deregulated by BRCA mutations, 24 were modulated by therapies. In particular, BIRC5, SIRT1, MYC, EZH2, and CSN2, are influenced by BRCA1, while BCL2, BAX, and BRIP1 by BRCA2 mutation. Moreover, the study evaluated the efficacy of several promising therapies, targeting only BRCA1/BRCA2-mutated cells. In this context, CDDO-Imidazolide was shown to increase ROS levels and induce DNA damage. Similarly, resveratrol decreased the expression of anti-apoptotic gene BIRC5 while increased SIRT1 both in vitro and in vivo. Other specific drugs were found to induce apoptosis selectively in BRCA-mutated cells, or block cell growth when the mutation occurs, i.e. 3-deazaneplanocin A, genistein or daidzein, and PARP inhibitors. Finally, over-representation analysis of the genes highlights ferroptosis and proteoglycan pathways as potential drug targets for more effective treatments”.
- The explanation of triple-negative breast cancer (TNBC) subtypes could benefit from a more concise presentation. A few claims are not adequately supported by recent citations (e.g., TNBC classification based on Lehmann).
Answer: The reviewer suggestion is appropriate; the Lehmann classification is dated. We reduced the TNBCs explanation and updated the paragraph, replacing some references.
Line 49: “Four molecular phenotypes (luminal A, luminal B, HER2+, and triple-negative breast cancer (TNBC)) are characterized by the presence of estrogen (ER+) and progesterone receptor (PR+), as well as human epidermal growth factor receptor 2 (HER2+/neu) [4]. TNBCs, characterized by worst prognosis, are the less frequently diagnosed BC (10-20%) [1]. They were previously divided into four subgroups [5], further categorized by whole-genome sequencing [6] or transcriptomic analysis [7].”
- The inclusion criteria for article selection are overly broad and should be refined for reproducibility. Additionally, the use of only 15 articles out of 305 raises concerns about selection bias. The reference to WebGestalt should include the exact version and access date for reproducibility.
Answer: As suggested by the reviewer, we refined the query, and now the number of articles decreases to 282. However, it remains elevated. We made other attempts to try other “keys” and formula, but proceeding in this way, we lose important articles containing some inclusion criteria that were essential for us, such as the connections among inclusion of a treatments in relationship to one/more gene/genes, which are modulated by the BRCA mutations, necessary to reach the goal of this review. Further, two papers concerning single-cell analysis were included in the list.
We have modified and tried to clarify this aspect better:
Line 184: “In order to analyze the most recent and promising literature, we generated a query based on a combination of the above inclusion criteria: (("genes, BRCA1"[MeSH Terms] OR " BRCA2 siRNA"[All Fields] AND "breast cancer"[All Fields]) AND ("siRNA"[All Fields] OR "DNA damage"[All Fields] OR "apoptosis"[All Fields] OR "deficiency"[All Fields] OR "inhibition"[All Fields] OR "expression of genes"[All Fields] OR "resistance"[All Fields]) NOT (Review[pt]) NOT (Meta-analysis[pt]) NOT ("Systematic Review"[pt])). This has re-turned 282 articles, a high number, which, however, guaranteed us to maintain that important inclusion criteria, essential for us, were not lost, mainly the connections among treatments in relationship to one/more gene/genes, which are modulated by the BRCA mutations, necessary to reach the goal of this review. After a manual curation of the literature, only 14 manuscripts were considered useful for further insights, due to the characteristics required.”
Answer: Concerning WebGestalt, we added the version and access date.
Line 219: “…. (WebGestalt 2024, https://www.webgestalt.org/ accessed on 20th December 2024).
- The results section occasionally lacks quantitative rigor, especially in describing drug efficacy and gene expression analysis. The pathway analysis is discussed but not linked clearly to clinical outcomes. Citations for pathway analysis should include primary sources from KEGG and related studies.
Answer: Reviewer has raised a valid point. Data derived from our analysis are focused on altered genes by BRCA mutations, also modulated by treatments (summarized in the new Figure 3), with the aim to highlight their relevance in the perspective of future applications, because they are currently carried out in cell culture or mouse models. For this reason, these findings cannot be associated with clinical outcomes yet. However, we have modified the Figure and included a new paragraph:
Line 710: “Our analysis evidenced relevant manuscripts that allow us to create interconnections focused on the deregulated genes by BRCA mutations and further modulated upon treatments [50,118,134-145]. In this context, experiments carried out in cell cultures or animal models are schematized in Figure 3 and further explored in the following paragraphs.”
Line 791: “Indeed, the combined therapy of talazoparib, a PARPi useful to affect BRCA1/2-mutated HER2- BC [149], and BETi is under evaluation in Phase II clinical trials to reduce solid tumors difficult to treat that could easily metastasize [150].”
Answer: Concerning pathway analysis, we underline the relevance of ferroptosis in relationship to prognostic risk, and targeting ferroptosis related genes could improve the sensitivity of other types of cancer. While in the proteoglycan pathway, we underline the role of decorin. We added, as follows:
Line 1007: “Currently, the ferroptosis pathway is acquiring relevance for its impact on patients' survival [221], especially ALB (included in our list) is reported as a prognostic factor [222], and induction of ferroptosis by SLC7A11 targeting sensitizes BRCA-mutated and PARPi resistant cancer [223] Ferroptosis induction could be an interesting strategy to overcome PARPi drug resistance in BC where this way could be activated, such as p53-positive tu-mors [220].”
Line 931: “From the therapeutic point of view, decorin expression strongly prevents processes associated with the spread of metastasis, also targetable by drugs, such as stachydrine [191].”
Finally, we have included in Supplementary File 1 the sources and the list of genes used for KEGG pathway analysis (Lines 222 and 880).
- Some sections are repetitive, particularly concerning synthetic lethality and PARP inhibitors. There is limited discussion on resistance mechanisms and future research directions. Some broad claims about therapy efficacy lack specific references.
Answer: Synthetic lethality is explained in the specific section” 4.2. Synthetic lethality: PARPi for BRCA-mutated tumors”. Other mentions to “synthetic lethality” are not always referred to PARPi but used to describe mechanisms of action of other drugs. Instead, concerning the PARPis resistance mechanisms, we better described this point, and we have included new references, as follows:
Line 471: “For instance, the reversion of BRCA mutations can remove frameshift mutation and restore the open reading frame [105]. This mechanism is clinically relevant after treatment with platinum. Gene reversion mutations can restore HR genes, both BRCA1/2 and RAD51C/D, to overcome the effect of the PARPi, such as a partially functional NHEJ mechanism [106]. Another mechanism involved the loss of PAR glycohydrolase with an increase in PAR chains. Additionally, resistance could depend on the inhibition of the degradation of the DNA replication fork performed by some nucleases [106]. In BRCA1/BRCA1-deficient cells, the levels of Enhancer of Zeste 2 Polycomb Repressive Complex 2 subunit (EZH2) and Pax transcription activation domain-interacting protein activity are decreased, affecting the recruitment of nucleases and protecting the fork. Indeed, the PARPi cytotoxic effect is countered by an upregulation of the ATR/Chk1 pathway, which results in enhanced replication fork stability. Furthermore, the loss of PARPi-sensitivity could be related to the presence of wild-type Phosphatase and tensin homolog, modifying cell growth and cell proliferation [107].
As described by Giudice et al. a murine model revealed another mechanism of PARPi resistance [108]. An overexpression of drug-efflux transporter genes (Abcb1a and Abcb1b) has been observed in BRCA1-deficient breast tumors, in which increased drug-efflux of PARPi, removing them from cells and, consequently, reducing their effects [108].”
Answer: Concerning the future perspective, we included some trials base on targeting of genes reported in Figure 3, with specific reference to PARPis.
Line 710: “Our analysis evidenced relevant manuscripts that allow us to create interconnections focused on the deregulated genes by BRCA mutations and further modulated upon treatments [50,118,134-145]. In this context, experiments carried out in cell cultures or animal models are schematized in Figure 3 and further explored in the following paragraphs.
These molecules can represent interesting therapeutic tools to fight this specific can-cer subtype. Data reporting IC50, dosage, and efficacy (when indicated) were summarized in Table 1.
Concerning PARP, although PARPi monotherapy produces a minimal impact on the overall survival of patients, it can enhance immunotherapy effectiveness.
KEYNOTE-162 (niraparib+pembrolizumab) and MEDIOLA (olaparib+durvalumab) trials explore PARPi-immunotherapy combinations, and JAVELIN (avelumab+talazoparib) trial demonstrated an acceptable safety profile [16]. However, further investigation is necessary.”
- The conclusion summarizes the key findings effectively, but it could be more forward-looking by suggesting specific research gaps and clinical implications.
We agree with the Reviewer that future perspectives are very important. Also, we have extensively modified the conclusions, as reported:
Line 1023: “Therapeutic advancements targeting the unique vulnerabilities of BRCA-mutated tumors, such as PARPi, represent considerable promise in both early and metastatic stages. However, drug resistance remains a significant obstacle. BET inhibitors (BETi) have emerged as promising agents exploiting synthetic lethality, particularly by triggering oxidative stress-related apoptosis through Myc and TXNIP production in BRCA1-deficient models. Moreover, DZNep treatment has selectively eliminated BRCA1-mutated cells by modifying EZH2 transcription, a crucial factor in DNA repair and cancer cell proliferation. In addi-tion to PARPi, chemopreventive agents like phytoestrogens have been investigated for their capacity to induce apoptosis and maintain genome stability, resulting in increased BRCA1/2 expression in BRCA2 knockdown cells. Another potential agent, CDDO, an imidazolide, can stimulate ROS production and DNA damage, inducing apoptosis in BRCA1 mutants. These findings emphasize the importance of a tailored treatment strategy, combining targeted therapies with immunotherapies or metabolic interventions to improve clinical outcomes. Despite significant progress that has been achieved, further research is essential to deepen the understanding of the molecular mechanisms involved and to develop innovative strategies that lead to more effective therapies for patients affected by BRCA-mutated BC. From this study, some interesting pathways suggest novel targetable processes. Ferroptosis has garnered attention as a novel therapeutic target. Its activation may overcome drug resistance and enhance the effectiveness of current treatments by targeting genes involved in oxidative stress and lipid peroxidation in those tumors in which it could be inducted. Also relevant is the modulation of proteoglycans in the extracellular space, where decorin could be induced, for instance, by stachydrine administration. We believe that the expanding role of precision medicine, including single cell sequencing and personalized will refine future therapeutic strategies and their applications of genetic testing to specific molecular subtyping.”
- Several references are outdated, particularly in the immunotherapy section. Ensure the inclusion of recent studies (2020–2024).
There are inconsistencies in formatting (e.g., missing DOIs and access dates).
Answer: Our comprehensive analysis is based on scientific articles published from 1997 to 2024. Nonetheless, to meet the reviewer's requests, we updated many citations, especially in the immunotherapy section, and we added access dates for websites or tools.
In our opinion, these are relevant:
- Salemme, V.; Centonze, G.; Cavallo, F.; Defilippi, P.; & Conti, L. The Crosstalk Between Tumor Cells and the Immune Microenvironment in Breast Cancer: Implications for Immunotherapy.Frontiers in oncology, 2021, 11. https://doi.org/10.3389/fonc.2021.610303
- De Santis, P.; Perrone, M.; Guarini, C.; Santoro, A. N.; Laface, C.; Carrozzo, D.; Oliva, G. R.; Fedele, P. Early-stage triple negative breast cancer: the therapeutic role of immunotherapy and the prognostic value of pathological complete response.Exploration of targeted anti-tumor therapy, 2024, 5, 232–250. https://doi.org/10.37349/etat.2024.00215
- Sriramulu, S.; Thoidingjam, S.; Speers, C.; & Nyati, S. Present and Future of Immunotherapy for Triple-Negative Breast Cancer.Cancers, 2024, 16, 3250. https://doi.org/10.3390/cancers16193250
- Yi, H.; Li, Y.; Tan, Y.; Fu, S.; Tang, F.; Deng, X. Immune Checkpoint Inhibition for Triple-Negative Breast Cancer: Current Landscape and Future Perspectives.Frontiers in oncology, 2021, 11, 648139. https://doi.org/10.3389/fonc.2021.648139
- Mittendorf, E. A.; Philips, A. V.; Meric-Bernstam, F.; Qiao, N.; Wu, Y.; Harrington, S.; Su, X.; Wang, Y.; Gonzalez-Angulo, A. M.; Akcakanat, A. et al. PD-L1 Expression in Triple-Negative Breast Cancer. Cancer Immunol Res 2014, 2, 361–370.
- Liu, B.; Zhou, H.; Tan, L.; Siu, K. T. H.; & Guan, X. Y. Exploring treatment options in cancer: Tumor treatment strategies. Signal transduction and targeted therapy, 2024, 9(1), 175. https://doi.org/10.1038/s41392-024-01856-7
Reviewer 2 Report
Comments and Suggestions for Authors
This review provides an in-depth analysis of the molecular pathways associated with BRCA1/BRCA2-mutated breast cancer and their potential as drug targets. Using network-based approaches, the authors identify key deregulated genes and pathways, such as BIRC5, SIRT1, MYC, and EZH2, that are modulated by BRCA mutations. The study highlights promising therapeutic agents, including CDDO-imidazolide, 3-deazaneplanocin A, phytoestrogens, resveratrol, and PARP inhibitors, that selectively target BRCA-mutant cells. Additionally, the role of ferroptosis and proteoglycan pathways in BRCA-mutated breast cancer is explored. Below are my suggestions to the authors:
1. The manuscript states that 15 out of 305 articles were selected, but it does not clearly explain why only these articles were considered useful. A more detailed justification for inclusion/exclusion criteria would strengthen this paper.
2. While the review discusses various drugs targeting BRCA mutations, it lacks comparative data on their effectiveness (e.g., IC50 values, clinical trial results). A table summarizing key findings from studies on these drugs would be useful.
3. The discussion is more focused on BRCA1, while BRCA2 is mentioned less extensively. Since BRCA2 mutations also lead to distinct biological consequences, a deeper comparison between BRCA1 and BRCA2-mutated tumors could enhance the review.
4. This manuscript highlights the promise of PARP inhibitors, but it does not sufficiently discuss mechanisms of resistance.
5. The Cytoscape-generated molecular networks are valuable, but they could be better annotated. The author may consider labeling the key genes, pathways, and interactions with simplified legends.
Author Response
Comments and Suggestions for Authors
Reply to Reviewer 2
This review provides an in-depth analysis of the molecular pathways associated with BRCA1/BRCA2-mutated breast cancer and their potential as drug targets. Using network-based approaches, the authors identify key deregulated genes and pathways, such as BIRC5, SIRT1, MYC, and EZH2, that are modulated by BRCA mutations. The study highlights promising therapeutic agents, including CDDO-imidazolide, 3-deazaneplanocin A, phytoestrogens, resveratrol, and PARP inhibitors, that selectively target BRCA-mutant cells. Additionally, the role of ferroptosis and proteoglycan pathways in BRCA-mutated breast cancer is explored. Below are my suggestions to the authors:
- The manuscript states that 15 out of 305 articles were selected, but it does not clearly explain why only these articles were considered useful. A more detailed justification for inclusion/exclusion criteria would strengthen this paper.
Answer: As suggested by the reviewer, we refined the query, and now the number of articles decreases to 282. However, it remains elevated. We made other attempts to try other “keys” and formula, but proceeding in this way, we lose important articles containing some inclusion criteria that were essential for us, such as the connections among inclusion of a treatments in relationship to one/more gene/genes, which are modulated by the BRCA mutations, necessary to reach the goal of this review. Further, two papers concerning single-cell analysis were included in the list.
We have modified and tried to clarify this aspect better:
Line 184: “In order to analyze the most recent and promising literature, we generated a query based on a combination of the above inclusion criteria: (("genes, BRCA1"[MeSH Terms] OR " BRCA2 siRNA"[All Fields] AND "breast cancer"[All Fields]) AND ("siRNA"[All Fields] OR "DNA damage"[All Fields] OR "apoptosis"[All Fields] OR "deficiency"[All Fields] OR "inhibition"[All Fields] OR "expression of genes"[All Fields] OR "resistance"[All Fields]) NOT (Review[pt]) NOT (Meta-analysis[pt]) NOT ("Systematic Review"[pt])). This has re-turned 282 articles, a high number, which, however, guaranteed us to maintain that important inclusion criteria, essential for us, were not lost, mainly the connections among treatments in relationship to one/more gene/genes, which are modulated by the BRCA mutations, necessary to reach the goal of this review. After a manual curation of the literature, only 14 manuscripts …”
- While the review discusses various drugs targeting BRCA mutations, it lacks comparative data on their effectiveness (e.g., IC50 values, clinical trial results). A table summarizing key findings from studies on these drugs would be useful.
Answer: As suggested by the reviewer we created a table summarizing the effectiveness of the drugs included in the network, and some trials are now mentioned.
Line 710: “Our analysis evidenced relevant manuscripts that allow us to create interconnections focused on the deregulated genes by BRCA mutations and further modulated upon treatments [50,118,134-145]. In this context, experiments carried out in cell cultures or animal models are schematized in Figure 3 and further explored in the following paragraphs.
These molecules can represent interesting therapeutic tools to fight this specific can-cer subtype. Data reporting IC50, dosage, and efficacy (when indicated) were summarized in Table 1.
Concerning PARP, although PARPi monotherapy produces a minimal impact on the overall survival of patients, it can enhance immunotherapy effectiveness.
KEYNOTE-162 (niraparib+pembrolizumab) and MEDIOLA (olaparib+durvalumab) trials explore PARPi-immunotherapy combinations, and JAVELIN (avelumab+talazoparib) trial demonstrated an acceptable safety profile [16]. However, further investigation is necessary.”
- The discussion is more focused on BRCA1, while BRCA2 is mentioned less extensively. Since BRCA2 mutations also lead to distinct biological consequences, a deeper comparison between BRCA1 and BRCA2-mutated tumors could enhance the review.
Answer: We understand the request, however most of the articles were focused on either BRCA1 or both, while less on BRCA2 mutated models. Some aspects are discussed as follows:
Line 172: “Around 80% of BRCA1 mutation carriers develop BC exhibit TNBC subtype. While germline PV mutations in BRCA2 are rarely associated with TNBC (2-16%) but are more commonly reported in association with ER+ BC [32].”
Line 263: “BRCA1 leads to defective DNA damage repair, abnormal centrosome duplication, G2/M cell cycle checkpoint defect, growth retardation, increased apoptosis, genetic instability and tumorigenesis, whereas the role of BRCA2 in cell cycle regulation is not fully under-stood, although evidence suggests its involvement [38,39,40].”
Line 330: “The BRCA2 role is less clearly defined than that of BRCA1. However, some evidence suggests a connection between BRCA2 and other transcription-related proteins, such as Smad3. Both Smad3 and BRCA2 contain the transcription-activation domains and exhibit functional and physical interactions. Previous studies have demonstrated that BRCA2 forms a complex with Smad3, both in vitro and in vivo, working synergistically to regulate gene transcription [61].”
The difference between BRCA1 and BRCA2 mutations has been reported in section 3.3, where their mutation has been linked to cell cycle dysregulation.
- This manuscript highlights the promise of PARP inhibitors, but it does not sufficiently discuss mechanisms of resistance.
Answer: As requested by the Reviewer, we better described this point, and we have included new references, as follows:
Line 471: “For instance, the reversion of BRCA mutations can remove frameshift mutation and restore the open reading frame [105]. This mechanism is clinically relevant after treatment with platinum. Gene reversion mutations can restore HR genes, both BRCA1/2 and RAD51C/D, to overcome the effect of the PARPi, such as a partially functional NHEJ mechanism [106]. Another mechanism involved the loss of PAR glycohydrolase with an increase in PAR chains. Additionally, resistance could depend on the inhibition of the degradation of the DNA replication fork performed by some nucleases [106]. In BRCA1/BRCA1-deficient cells, the levels of Enhancer of Zeste 2 Polycomb Repressive Complex 2 subunit (EZH2) and Pax transcription activation domain-interacting protein activity are decreased, affecting the recruitment of nucleases and protecting the fork. Indeed, the PARPi cytotoxic effect is countered by an upregulation of the ATR/Chk1 pathway, which results in enhanced replication fork stability. Furthermore, the loss of PARPi-sensitivity could be related to the presence of wild-type Phosphatase and tensin homolog, modifying cell growth and cell proliferation [107].
As described by Giudice et al. a murine model revealed another mechanism of PARPi resistance [108]. An overexpression of drug-efflux transporter genes (Abcb1a and Abcb1b) has been observed in BRCA1-deficient breast tumors, in which increased drug-efflux of PARPi, removing them from cells and, consequently, reducing their effects [108]”.
As described by Giudice et al. a murine model revealed another mechanism of PARPis resistance [ref]. An overexpression of drug-efflux transporter genes (Abcb1a and Abcb1b) has been observed in BRCA1-deficient breast tumors, in which increased drug-efflux of PARPis, removing them from cells and, consequently, reducing their effects [ref].”
- The Cytoscape-generated molecular networks are valuable, but they could be better annotated. The author may consider labeling the key genes, pathways, and interactions with simplified legends.
Answer: As requested by the reviewer, we modified Figure (now figure 3), highlighting “singularly” on the bottom the crucial networks.
Reviewer 3 Report
Comments and Suggestions for Authors
The manuscript titled “Druggable Molecular Networks in BRCA1/BRCA2-Mutated Breast Cancer” provides a comprehensive analysis of the complex relationship between BRCA1/2 mutations, gene expression changes, and the underlying biological mechanisms driving breast cancer development and treatment responses. Through a thorough review of the literature, the authors focused on deregulated genes influenced by BRCA1 and BRCA2 mutations and their modulation by therapeutic interventions. By constructing molecular networks and identifying key pathways, the study offers valuable insights into potential targets for future precision therapies. Additionally, it highlights promising drugs capable of inducing apoptosis in cancer cells, either by selectively eliminating BRCA-mutated cells or inhibiting their proliferation. The paper is well-written and presents a significant genetic analysis of BRCA-driven breast cancer, making it a valuable contribution to the field. With minor revisions, it is suitable for publication in Biology Journal. Please consider the following comments and recommendations:
1) The introduction is overly lengthy. Please condense it for clarity and conciseness.
2) Add a theoretical framework in a conceptual figure.
3) Line 35: Please what do you mean here by this “The non-invasive type of BC, known as in situ, can be identified earlier and remains localized to the ducts or lobules of the mammary gland, thus is.” Enhance this for clarity
4) Line 42: Additionally, from a molecular perspective, BC can be differentiated into four phenotypes based on the presence of estrogen (ER+) and progesterone receptor (PR+), as well as human epidermal growth factor receptor 2 (HER2+/neu). These four BC types are identified as luminal A, luminal B, HER2+, and triple-negative breast cancer (TNBC). Please add reference here.
5) Please add the mechanism of action of PARPi
6) There are occasional typographical errors. For example, in Line 123: Leading to his rolo of “guardian of genome”.
7) Please mention limitations, study weakness and strengths in the end of discussion part.
8) Please add a section about challenges in drug development and resistance mechanisms before the conclusion.
9) Add study novelty at the end of the introduction section
Comments on the Quality of English Language
Minor English editing is required.
Author Response
Comments and Suggestions for Authors
Reply to Reviewer 3
The manuscript titled “Druggable Molecular Networks in BRCA1/BRCA2-Mutated Breast Cancer” provides a comprehensive analysis of the complex relationship between BRCA1/2 mutations, gene expression changes, and the underlying biological mechanisms driving breast cancer development and treatment responses. Through a thorough review of the literature, the authors focused on deregulated genes influenced by BRCA1 and BRCA2 mutations and their modulation by therapeutic interventions. By constructing molecular networks and identifying key pathways, the study offers valuable insights into potential targets for future precision therapies. Additionally, it highlights promising drugs capable of inducing apoptosis in cancer cells, either by selectively eliminating BRCA-mutated cells or inhibiting their proliferation. The paper is well-written and presents a significant genetic analysis of BRCA-driven breast cancer, making it a valuable contribution to the field. With minor revisions, it is suitable for publication in Biology Journal. Please consider the following comments and recommendations:
1) The introduction is overly lengthy. Please condense it for clarity and conciseness.
Answer: We tried to shorten some points, but we were unable to shorten them further to address issues raised by other reviewers.
2) Add a theoretical framework in a conceptual figure.
Answer: We have described the framework in a new figure in the Materials and Methods section (Line: 206).
3) Line 35: Please what do you mean here by this “The non-invasive type of BC, known as in situ, can be identified earlier and remains localized to the ducts or lobules of the mammary gland, thus is.” Enhance this for clarity
Answer: We better explain the concept by adding a reference.
Line 38: “There are three distinct types of breast neoplasia. Ductal carcinoma in situ (DCIS) arises in epithelial cells forming the breast ducts. Several studies suggest that at least one third of DCIS cases will eventually progress to invasive cancer if left untreated. Lobular carcinoma in situ (LCIS) develops in milk producing tissue (the functional part of the breast gland), conferring an increased risk of developing invasive cancer. Cancer is considered invasive when the tumoral cells leak the basal membrane and infiltrate the surroundings. However, most BC are invasive or infiltrating, and the prognosis is dependent on the stage of the disease [2].”
4) Line 42: Additionally, from a molecular perspective, BC can be differentiated into four phenotypes based on the presence of estrogen (ER+) and progesterone receptor (PR+), as well as human epidermal growth factor receptor 2 (HER2+/neu). These four BC types are identified as luminal A, luminal B, HER2+, and triple-negative breast cancer (TNBC). Please add reference here.
Answer: The reviewer suggestion is appropriate. We add the reference related to the paragraph indicated above.
Line 52: Orrantia-Borunda, E.; Anchondo-Nuñez, P.; Acuña-Aguilar, L. E.; Gómez-Valles, F. O.; Ramírez-Valdespino, C. A. Subtypes of breast cancer. In Breast Cancer; Mayrovitz, H. N., Ed.; Exon Publications: Brisbane (AU), 2022, Chapter 3.
5) Please add the mechanism of action of PARPi
Answer: As suggested by the reviewer, we added a short description of PARPi mechanism of action, to improve the comprehension of the paragraph.
Line 460: “PARPi mechanism of action involves PARP-1 protein binding at a single-stranded break or DNA lesion, interrupting its catalytic cycle, resulting in replication fork advancement, inducing double-strand breaks [96]. The ability of PARPi to bind PARP-1 NAD+ binding site provokes conformational changes, leading to the stabilization of inter-action between PARP-1 and DNA, preventing the catalytic site from reverting to the inactive state. In this manner, PARP-1 lacks its standard function [97]. The accumulation of SSBs causes a stop of replication fork advancement.”
6) There are occasional typographical errors. For example, in Line 123: Leading to his rolo of “guardian of genome”.
Answer: We thank the Reviewer for pinpointing the typos, we proceeded to have them fixed in the main text.
7) Please mention limitations, study weakness and strengths in the end of discussion part.
Answer: We have added a new paragraph highlighting the limitation of the study.
Line 1023: “Therapeutic advancements targeting the unique vulnerabilities of BRCA-mutated tumors, such as PARPi, represent considerable promise in both early and metastatic stages. However, drug resistance remains a significant obstacle. BET inhibitors (BETi) have emerged as promising agents exploiting synthetic lethality, particularly by triggering oxidative stress-related apoptosis through Myc and TXNIP production in BRCA1-deficient models. Moreover, DZNep treatment has selectively eliminated BRCA1-mutated cells by modifying EZH2 transcription, a crucial factor in DNA repair and cancer cell proliferation. In addition to PARPi, chemopreventive agents like phytoestrogens have been investigated for their capacity to induce apoptosis and maintain genome stability, resulting in increased BRCA1/2 expression in BRCA2 knockdown cells. Another potential agent, CDDO, an imidazolide, can stimulate ROS production and DNA damage, inducing apoptosis in BRCA1 mutants. These findings emphasize the importance of a tailored treatment strategy, combining targeted therapies with immunotherapies or metabolic interventions to improve clinical outcomes. Despite significant progress that has been achieved, further research is essential to deepen the understanding of the molecular mechanisms involved and to develop innovative strategies that lead to more effective therapies for patients affected by BRCA-mutated BC. From this study, some interesting pathways suggest novel targetable processes. Ferroptosis has garnered attention as a novel therapeutic target. Its activation may overcome drug resistance and enhance the effectiveness of current treatments by targeting genes involved in oxidative stress and lipid peroxidation in those tumors in which it could be inducted. Also relevant is the modulation of proteoglycans in the extracellular space, where decorin could be induced, for instance, by stachydrine administration. We believe that the expanding role of precision medicine, including single cell sequencing and personalized will refine future therapeutic strategies and their applications of genetic testing to specific molecular subtyping.”
8) Please add a section about challenges in drug development and resistance mechanisms before the conclusion.
Answer: We added the following pargraphs:
Line 688: “Indeed, the BRCA-mutated BC exhibits specific clinical characteristics, such as increased sensitivity to PARPi, DNA-damaging agents such as platinum-based chemotherapies, and potentially decreased sensitivity to CDK4/6 inhibitors. Prompt determination of BRCA status is crucial to establish effective treatment strategies for patients.
Targeted drugs, immunotherapies, and combined treatments are being developed to maximize effectiveness and minimize side effects, providing new hope for BRCA-mutated patients. Research on novel therapies for malignancies linked to BRCA mutations is a fast-moving field that aims to improve patient prognosis and quality of life. Individuals carrying BRCA1/2 mutations face an increased risk of developing breast, ovarian, prostate, and pancreatic cancers. Consequently, therapeutic strategies that exploit the specific abnormalities of cancer cells, also applicable to hereditary breast cancer, allow for the utilization of PARPi, as previously mentioned [2].
Despite the potential of platinum-based chemotherapies, PARPi, and CDK4/6 inhibitors, a large number of patients develop resistance to treatment or toxicity, which represents the most important problem [2].”
Line 703: “This subtype of BC remains a big challenge. Advancements in this field are progressing rapidly with several potential new avenues emerging. Although PARPi potential has been largely explored, the resistance to this treatment is a big issue. Currently, the combinations of conventionally used BC therapies together with new molecules are under investigation, to improve patient outcomes. Likewise, in PARPi treatments, synthetic lethality has been employed by other molecules, such as BETi [134].”
9) Add study novelty at the end of the introduction section
Answer: We further highlighted the importance of our study, adding the following paragraph to the end of Materials and Methods, including it in this position because it seemed to us to be a more logical sequence based on the division of the paragraphs.
Line 224: “The resulting modulation of the involved genes could represent a valuable resource to explore new targets for drug design and repurposing as chemopreventive agents. Hence, our investigation shed light on potential target genes, which have already been shown to be affected by molecules and compounds.
This review aims to analyze the literature, based on BRCA-mutated models, mostly to verify which genes are modulated, up or down-regulated, by the BRCA gene mutation. In this perspective, we only selected the studies including all our criteria, discovering several genes that are not commonly associated with BRCA mutation in literature. Furthermore, our analysis identified genes regulated by treatment and compounds used in combination therapies. Despite the lack of treatment, we included two single-cell-based studies that highlighted not only some key genes but also features such as the induction of basal-mesenchymal transition of luminal progenitors, responsible for breast cancer development in mutation carriers. Besides that, luminal progenitors have also been investigated for their ability to develop alveolar differentiation during premalignancy in BRCA1 mutated models.”
Round 2
Reviewer 1 Report
Comments and Suggestions for Authors
All comments are justified and satisfactory. No further comments on the scientific and technical aspects of the manuscript.
Reviewer 2 Report
Comments and Suggestions for Authors
The author answered all my questions. The manuscript looks ok overall